# Using Machine Learning and Software-Defined Networking to Detect and Mitigate DDoS Attacks in Fiber-Optic Networks

Sulaiman Alwabisi [1], Ridha Ouni [1,*] and Kashif Saleem [2]

1 Department of Computer Engineering, College of Computer and Information Sciences (CCIS), King Saud University, Riyadh 11461, Saudi Arabia
2 Center of Excellence in Information Assurance (CoEIA), King Saud University, Riyadh 12372, Saudi Arabia
* Correspondence: rouni@ksu.edu.sa

**Abstract:** Fiber optic networks (FONs) are considered the backbone of telecom companies worldwide. However, the network elements of FONs are scattered over a wide area and managed through a centralized controller based on intelligent devices and the internet of things (IoT), with actuators used to perform specific tasks at remote locations. During the COVID-19 pandemic, many telecom companies advised their employees to manage the network using the public internet (e.g., working from home while connected to an IoT network). Theses IoT devices mostly have weak security algorithms that are easily taken-over by hackers, and therefore can generate Distributed Denial of Service (DDoS) attacks in FONs. A DDoS attack is one of the most severe cyberattack types, and can negatively affect the stability and quality of managing networks. Nowadays, software-defined networks (SDN) constitute a new approach that simplifies how the network can be managed through a centralized controller. Moreover, machine learning algorithms allow the detection of incoming malicious traffic with high accuracy. Therefore, combining SDN and ML approaches can lead to detecting and stopping DDoS attacks quickly and efficiently, especially compared to traditional methods. In this paper, we evaluated six ML models: Logistic Regression, K-Nearest Neighbor, Support Vector Machine, Naive Bayes, Decision Tree, and Random Forest. The accuracy reached 100% while detecting DDoS attacks in FON with two approaches: (1) using SVM with three features (SOS, SSIP, and RPF) and (2) using Random Forest with five features (SOS, SSIP, RPF, SDFP, and SDFB). The training time for the first approach was 14.3 s, whereas the second approach only requires 0.18 s; hence, the second approach was utilized for deployment.

**Keywords:** fiber optic networks; software defined networking; machine learning; distributed denial of service; internet of things

## 1. Introduction

Currently, telecom fiber optic networks (FON) make up the backbone of most worldwide telecom internet service providers (ISPs). Telecom networks enable analog or digital information to be communicated between different sites using electromagnetic or optical signals, supported by fiber optic-based 4G and 5G networks [1]. However, telecom networks are subject to distributed denial of service (DDoS) attacks (i.e., hacking); such cyberattacks cause telecom users to suffer denied or delayed telecom services. According to Corero Network Security [2], a distributed denial-of-service (DDoS) protection and mitigation provider, organizations worldwide have independently experienced an average of 270 DDoS attacks per month, averaging nine attacks daily, a 13% increase from 2019.

Corero Network Security elaborates that 86% of cyberattacks happen less than ten minutes apart. Even a few minutes of network or server downtime can prove costly for organizations in terms of lost revenue, reduced customer confidence, negative impacts on network continuities, and most importantly, the overall consequences of reputation damage. Therefore, protecting fiber optic telecom networks and ensuring business continuity requires organizations to embed a traffic classification recognition system within their

network functions and management systems that can instantaneously identify different virtual traffic flow inputs and types and alert network operators when a network or server occurrence is suspicious.

Traffic classification programs allow network operators to manage various services and allocate network resources more efficiently. Widely used traffic classification techniques include the port-based approach, deep packet inspection (DPI), and autonomous machine learning. The port-based method uses transmission control protocol (TCP) and user datagram program (UDP) port numbers to determine the incoming and outgoing application or traffic types. On the one hand, modern applications use well-known ports, such as TCP port 80, for hypertext transfer protocols (HTTP). On the other hand, most modernized applications operate on dynamic ports, which makes the port-based approach no longer efficient for organizations [3].

DPI compares the payload of traffic flows with predefined patterns embedded within program databases, allowing systems to identify applications or locations using common expressions from which traffic flows originate. Moreover, a DPI-based approach is noted to have high classification accuracy. However, drawbacks of utilizing the DPI-based approach are that DPI can only identify applications for which patterns are available. Another concern with respect to DPI is the high computational cost of managing, monitoring, and keeping the network online, as all traffic flows must be checked and DPI cannot classify encrypted traffic on the internet. In light of these drawbacks of DPI, ML-based techniques have been extensively studied for network security, as many traffic flows must be collected first in order for ML techniques to be applied to extract knowledge from the steady traffic flow. In this context, the most popular and efficient ML algorithms are Logistic Regression, K-Nearest Neighbors, Support Vector Machine, Naive Bayes, Decision Tree, and Random Forest. These ML algorithms have demonstrated high performance in different types of applications [3–5].

Software-defined networking (SDN) is another emerging concept that intends to replace traditional networks by dismantling vertical integration by separating the network's control logic from the underlying switches and routers. This implies that logical network control is more centralized, and allows the network to program itself extensively [6] using ML-based approaches. Many vendors predict that IoT can help with SDN decisionmaking in managing networks [7], for example, telecom network management of connectivity from IoT devices to the cloud. These networks are traditionally built using physical equipment, and need to ensure adequate uptime for services. As service providers start to deploy SDN for IoT in their networks, the front-end challenges are to maintain the level of reliability and security that customers expect. Furthermore, SDN can assist in driving the expansion of IoT-enabled devices.

DDoS attacks are toxic to any network. Different techniques can be used to attack a specific network, including ICMP flood, TCP SYN flood, and UDP flood, which are harmful to FON. DDoS attacks can harm the physical components of telecom equipment as well. Organizations can efficiently utilize ML and SDN capabilities to detect and mitigate DDoS attacks in FON to overcome network security issues [6]. However, organizations can use multiple ML approaches to appropriate and resolve network security concerns.

Selecting a specific model is challenging due to various aspects, including the format of the dataset (numerical, categorical, graphical, or textual), nature of the problem (classification, regression, clustering), and quantity of features in the dataset (instances, dependent and independent variables). Here, the performance evaluations achieved in recent related works have been considered to select the most appropriate ML models.

This paper aims to evaluate the six most well known ML algorithms (Logistic Regression, K-Nearest Neighbor, Support Vector Machine, Naive Bayes, Decision Tree, and Random Forest) for protecting FON from DDoS attacks. Our performance evaluation is based on classification metrics (Accuracy, Precision, Recall, and F1 score), and further contributes to presenting a comprehensive comparison of ML models. The ML model

providing the highest performance is then implemented in FONs to improve the network efficiency in terms of throughput and delay.

The rest of this paper is organized as follows. Section 2 covers the necessary background, and Section 3 reviews published works in the field of detecting and mitigating DDoS attacks in FONs. Then, the problem and the research motivation is described in Section 4. Section 5 introduces the proposed methodology. Section 6 presents the obtained results and discusses them, along with the limitations of this research and suggestions for future work. Finally, the primary findings and conclusions of this research are presented in Section 7.

## 2. FON, SDN, and ML Interaction for Security Applications

With the help of SDN and ML algorithms, the proposed work is concerned with applying DDoS attack detection and mitigation for FONs. Figure 1 depicts the topics related to our research work.

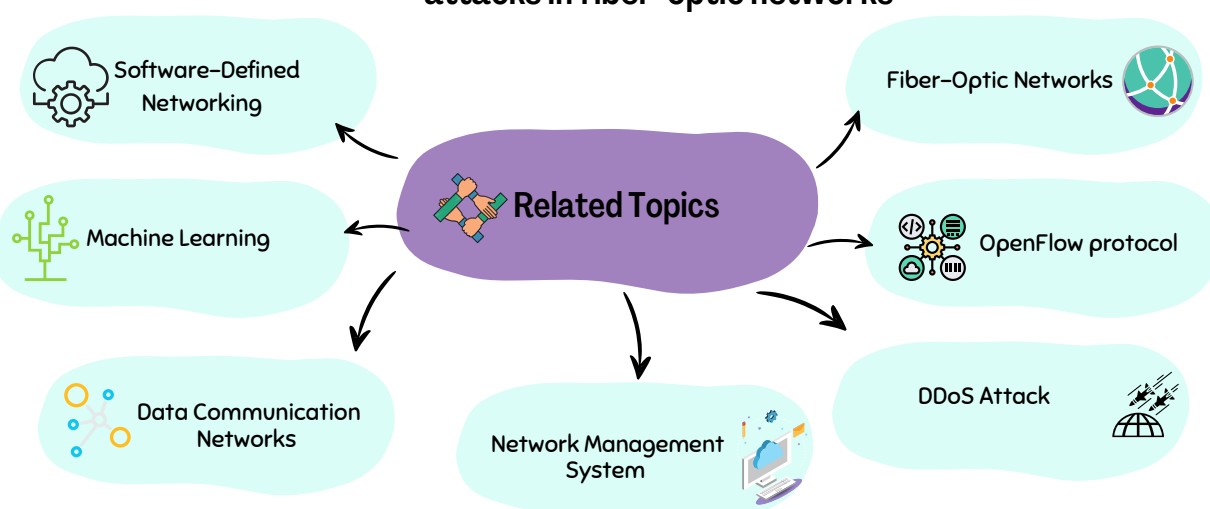

**Figure 1.** Related topics.

### 2.1. Fiber Optical Networks

Fiber Optical networks (FON) play a crucial role in developing high-quality and high-speed telecommunication systems. Today, optical fibers are used in telecommunications links, the Internet, and local area networks (LAN) to achieve high signaling rates [8]. Synchronous Digital Hierarchy (SDH) is a well-known technology for data communication and telecom networks, and is used for high-speed data transmission and to deliver digital signals of varying capacities [9].

### 2.2. Network Management System

A network management system (NMS) is an application or collection of tools that allows network engineers to manage, configure, and monitor the various components of a network within a stronger network management framework. Figure 2 shows the NMS system, which is often located in the Network Operations Center (NOC).

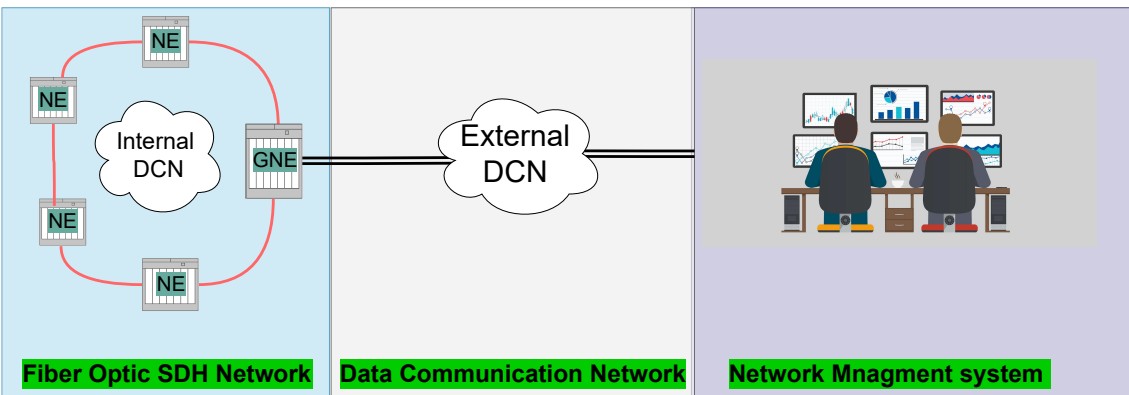

**Figure 2.** Fiber optic network architecture.

### 2.3. Data Communication Network

A data communication network (DCN) is a network in which operation, administration, and maintenance (OAM) information is shared between network elements (NEs) in the network management system (NMS). It ensures that systems and managed devices can talk to each other [10]. The DCN can be external or internal. In Figure 2, for example, an external DCN connects the NMS to Gateway Network elements (GNE) and an internal DCN lets NEs exchange information about OAM.

### 2.4. Data Communication Channel

The Data Communication Channel (DCC) feature utilizes the SDH Operation Administration and Maintenance (OAM) channel to manage devices that support SDH interfaces. The SDH standard supports a wide range of capabilities for operations, administration, management, and provisioning (OAMP) [11]. Figure 3 shows a basic SONET STM-1 frame comprising nine rows and ninety columns [12]. According to standards, the OAM channels that transport management information, alerts, and management commands are as follows:

- Overhead for the Generation Section ranges from D1 to D3 bytes.
- Bytes D4 through D12 are taken up by the Multiplex Section overhead.

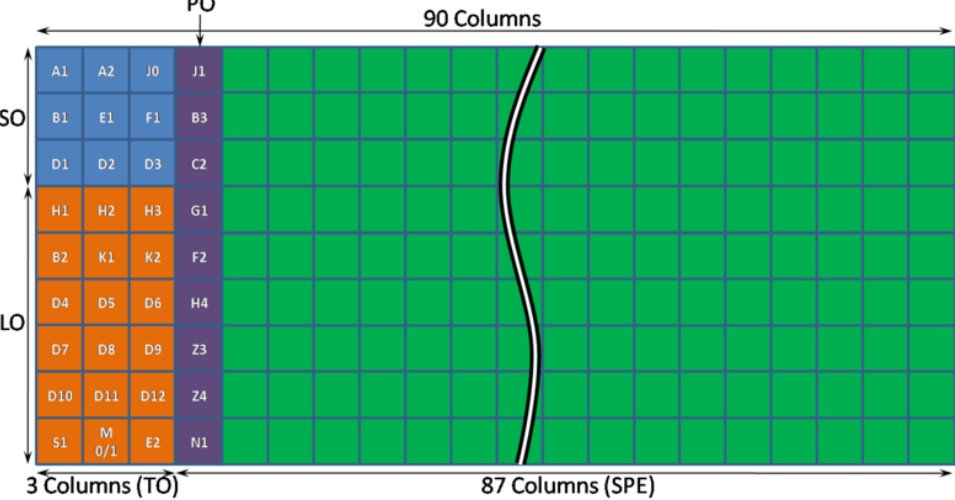

**Figure 3.** SDH frame structure [13].

The Data Communication Channel (DCC) refers to these extra bytes. The line-level DCC is a 576-kbps OAM channel, and the section-level DCC is a 192-kbps OAM channel [14].

### 2.5. Normal and Attack Traffic in DCNs of Fiber Optic Networks

Traffic in data communication networks of fiber optic networks can be categorized into two types, namely, normal and attack traffic.

#### 2.5.1. Normal Traffic

Normal traffic in the DCN of a Fiber-optic network can be described as the daily management activity, such as operation/maintenance activity, needed to ensure that all the fiber optic network elements are monitored and controlled at all times using NMS applications. Examples of normal traffic are File Transfer Protocol (FTP), Hypertext Transfer Protocol (HTTP), Simple Network Management Protocol (SNMP), and Internet Control Message Protocol (ICMP), which are discussed in the following paragraphs.

1. **FTP** is a protocol operating at the application layer to ensure files can be sent between computers using TCP connections [15]. FTP can be used for Backup/Restore capability between servers and nodes in fiber-optic networks and to simulate FTP traffic in the DCN. The Distributed Internet Traffic Generator (DITG) tool described in Section 5 can be used at this level.
2. **HTTP** is a protocol used to request and dispense web content based on plain text. Many websites employ HTTPS to encrypt traffic with Transport Layer Security (TLS), making the internet more secure [16]. NMS applications use HTTP or HTTPS to display a node's graphical user interface (GUI) and monitor its operation. Section 5 describes the Curl tool used to simulate HTTP traffic in the DCN.
3. **SNMP** is an internet standard used for managing devices (switches, servers, workstations, printers, routers, and telecom equipment) on IP networks. Most NMSs use SNMP to remotely monitor, set up, change, and fix networks. SNMP uses UDP port 161 to send and receive requests and port 162 to receive traps from managed devices [15]. As described in Section 5, the SNMP trap tool is used to simulate SNMP traffic in the DCN.
4. **ICMP** is a protocol used to verify the connections between various network components and determine whether data has reached the destination [17]. ICMP is a crucial component of error reporting and network transmission testing. However, distributed denial-of-service (DDoS) attacks can be carried out using ICMP. The Ping and Hping3 tools are used to simulate ICMP traffic in the DCN, as described in Section 5.

#### 2.5.2. Attack Traffic

One attack traffic type in the DCN of an FON is a DDoS attack, which constitutes a risk to any network. A DDoS attack aims to prevent genuine users from using a system or network resource, and works by flooding the target from many different directions. The following methods can be used to flood a target system:

1. **SYN Flood** is an attack targeting the victim's machine by starting a TCP connection. Because of this, the victim receives a large number of SYN packets, while no ACK is sent back to the victim. This utilizes many resources on the victim's machine and prevents legitimate users from being served [18].
2. **ICMP Flood** is an attack that exhausts all of the victim's resources by flooding the server with pings (echo requests) that keep it busy sending echo replies [18].
3. **UDP Flood** is an attack that attempts to take down servers by flooding the targeted host with many UDP packets to random ports. Often, attackers use UDP's connectionless functionality to broadcast a continuous stream of UDP data packets to the victim's workstation [18].
4. **HTTP Flood** is an attack that occurs at the application layer and targets web servers and apps. Typically, the attacker sends many HTTP GET and POST requests to a webserver [19]. A tool called Hping3, which is described in Section 5, can be used to simulate all the DDoS attacks mentioned above.

*2.6. Software-Defined Networking*

A software-defined network (SDN) is a type of network architecture that allows the hardware parts to be built and designed virtually [20]. Figure 4 shows the three layers comprising the architecture of a software-defined network [21,22]. These layers can be explained in the following way:

- **Application Layer:** a layer that contains applications and programs as well as services such as load balancing, quality-of-service, and a firewall.
- **Control layer:** a central controller that manages the network traffic. It uses the OpenFlow protocol to communicate with the infrastructure layer to monitor and control the entire network.
- **Infrastructure layer:** this layer has both physical and virtual network forwarding hardware devices that use OpenFlow protocols, such routers, switches, and access points.

**Figure 4.** SDN architecture [20].

*2.7. OpenFlow Protocol*

The OpenFlow protocol is at the heart of SDN technology, and an SDN switch with an OpenFlow switch promises communication networks that are flexible and easy to set up [23]. OpenFlow is an SDN-compatible programmable network protocol. It is used by OpenFlow switches and controllers to communicate with one another. OpenFlow decouples network device programming from the underlying hardware and provides a standard method for delivering a centralized and programmable network that can swiftly adapt to changing network needs. The OpenFlow switch is an OpenFlow-enabled data switch that communicates with an external controller over the OpenFlow channel.

*2.8. Machine Learning*

Machine learning enables systems deploying artificial intelligence algorithms to perform the required tasks effectively. Statistical learning methods are the core of AI systems

that make machines more intelligent. A method based on machine learning usually constitutes an approach with two main parts, a training phase and a decisionmaking phase. First, a training dataset is used to allow the system model to learn various features during the training phase. Then, the system can use the trained model to determine the estimated output for each new input [24,25].

## 3. Related Works

Multiple studies have addressed how to detect DDoS attacks. Various identification techniques, such as machine learning (ML), are explored from the available scholarly literature within this review. Our investigation shows that DDoS attack detection methods using ML have been actively conducted and are essential to monitoring and resolution operations for organizations.

Mishra et al. [26] proposed that cloud computing security with ML can act as a mitigation technique against cyber threats. Their proposed SDN is a multilayer composition of ML with a self-defense system that effectively detects and mitigates cyberattacks to protect cloud-based enterprise solutions. The results showed the accuracy of the proposed ML techniques and their effectiveness in attack detection and mitigation processes.

Dennis et al. [18] used the University of California—Los Angeles (UCLA) dataset and then modified it by adding traffic flow entries of simulated traffic. Next, they used native OpenFlow counters, utilizing the number of packets, number of bytes, and flow duration as features to build their desired model. Then, a flow-based DDoS attack detection system was developed using a random forest (RF) and ML embedded algorithm with weighted voting. The intended approach was to send flow statistics collected from the switches to the RF classifier every 10 s. When the system confirmed an attack, a mitigation module was implemented, resulting in attack traffic ceasing and preventing further breakdown and infiltration of the switches in the network or systems.

Isaac et al. [27] proposed a solution to detect (DDoS) attacks within an SDN using a support vector machine (SVM) to classify network traffic as abnormal (i.e., suspicious) or normal. However, the proposed security application only outlined the manipulation of two significant DDoS attack types: internet protocol (IP) spoofing and synthesized transaction processing performance council (i.e., TCP SYN) flooding. On the other hand, Rahman et al. [28] evaluated ML techniques, such as the J48 (a form of Iterative Dichotomiser 3), RF, SVM, and K-nearest neighbors (K-NN) algorithms, to detect and stop DDoS attacks in an SDN network. Their evaluation method involved training, selecting the most suitable model for the proposed network, and applying the desired model in mitigation and prevention scripts within the network and systems. As result, Rahman et al. demonstrated that J48 outperforms the other ML algorithms, especially in the training and testing phases.

Khashab et al. [29] evaluated six different ML algorithms: Decision Tree (DT), Logistic Regression (LR), Naïve Bayes (NB), KNN, RM, and SVM. Their investigation showed that RF was the best-performing ML algorithm. Additionally, the results showed that their proposed RF model could detect attacks accurately and immediately with a low probability of stalling regular traffic.

The method proposed by Vishal Kumar in [30] is implemented using a Ryu controller and mininet network simulator with OpenFlow SDN protocol. The presented method integrates statistical and machine learning methods to efficiently detect and mitigate DDOS attacks in an SDN, accomplishing an accuracy of 99.26% and a detection rate of 100% in detecting and mitigating DDoS attacks.

Ahmed [31] proposed a novel DDoS detection system initiated by a semi-supervised algorithm with an LR classifier. The author explained that the algorithm is executed as software modeled by a Phyton-based open-source (i.e., POX) SDN controller, and compared various test scenarios with statistical approaches to prove that the proposed detection system has a better attack detection rate with a slower reaction time.

Similarly, Kotb et al. [32] designed and implemented a security guard model to solve DDoS attacks on POX SDN controllers. Their model, named SGuard, is represented as a novel five-tuple feature vector utilized for classifying traffic flow by employing SVM. Mininet was used to assess SGuard in network and software environments, allowing SGaurd to assess the system's performance in terms of delay, bandwidth, traffic flow, and accuracy.

Contrarily, Gadallah et al. [33] provided an ML DDoS technique using newly noted features for SDNs. The feature characteristics in this work were informed by a dataset that had input from a linear SVM classifier. The classifier then trained the model with the kernel radial basis function. The SVM model used algorithms such as DT, KNN, NB, and RF as comparative correlations. The mediation system created by Gadallah et al. conclusively prevented suspicious network and server activities, and the questionable information was stored for future implementations to ensure more accurate mediation. Furthermore, the experiment conducted by Gadallah et al. determined that their proposed technique could detect attacks with higher accuracy and produce lower false alarm rates than other related methods.

Mohammed et al. [34] proposed another ML DDoS mitigation technique for SDN similar to other works mentioned in this literature review. They created a model for DDoS detection in an SDN using the Network Security Laboratory (NSL) and Knowledge Discovery and Data (KDD) datasets. After training the model on these datasets, the authors used realistic DDoS attacks to assess their proposed model, showing that their proposed technique equates favorably to the current methods in terms of providing increased performance and accuracy.

Kyaw et al. [35] proposed a detector for DDOS attacks. The authors classified normal and attack traffic in the SDN network using ML algorithms and compared polynomial SVM to the existing linear SVM using scapy, a packet generation tool, and a component-based software package that defines networking frameworks (i.e., the Ryu SDN controller). According to their experimental results, the polynomial SVM achieved 3% better accuracy and a 34% decrease in the false alarm rate compared to the linear SVM.

A hybrid ML model was used by Deepa et al. [36] to protect an experimental controller from DDoS attacks. Experimental results showed that their hybrid ML model provided better accuracy, a higher detection rate, and a lower false alarm rate compared to simple ML models. On the other hand, Nurwarsito et al. [37] investigated a DDoS attack detection and mitigation system constructed based on SDN architecture using an RF ML algorithm. The RF algorithm classified normal and attack packets based on their associated flow entries. If the packets were classified as a DDoS attack, the attacks were mitigated by flow rules to the switch. Based on examinations conducted by the authors, this detection system could detect DDoS attacks with an average accuracy of 98.38% with an average detection time of 36 milliseconds (ms). The system was able to mitigate DDoS attacks with an average mitigation time of 1179 ms, which reduced the average number of attack packets that entered the victimized experimental host by up to 15,672. The concurrent reduction in the controller's intermediate central processing unit (CPU) usage equated to 44.9%.

To complete this literature review, Sudar [38] offered an ML technique to detect malicious traffic that implemented DT and SVM. Test outcomes showed that the DT and SVM algorithms generally provided better accuracy and detection rate. Finally, Ye et al. [39] used an SDN environment, mininet, and a floodlight within a simulation platform constructed of six-tuple characteristic values applied to the switch flow table. They built a DDoS attack model by combining SVM classification algorithms. Their experiments showed that the average accuracy rate of the proposed method was 95.24%, alongside the small amount of flow that was collected. Thus, this proposed work is highly informative with respect to detecting DDoS attacks in SDN. Table 1 summarizes the above literature review.

**Table 1.** Comparison of related work.

| Ref | Scope | ML Algorithm | Dataset | Feature | Accuracy | Limitations |
|-----|-------|--------------|---------|---------|----------|-------------|
| [18] | Detection/Mitigation | RF | UCLA dataset and Synthetic | Three feature | 97.70% | Only one ML model is being evaluated. |
| [26] | Detection/Mitigation | SVM | Synthetic | Five features | 98.52% | Only layer 3 feature collection duplicates dataset instances. |
| [27] | Detection | SVM | Synthetic | Five features | High | Only layer 3 feature collection duplicates dataset instances. Some features remain unchanged. |
| [28] | Detection/Mitigation | J48 RF SVM K-NN | Synthetic | 24 Features | J48 classifier is the best | High complexity due to many features. The detection time is high. |
| [29] | Detection | SVM LR KNN DT NB RF | Synthetic | Five features | 94.99% 98.90% 86.41% 99.11% 99.64% 99.76% | No mitigation. |
| [30] | Detection/Mitigation | SVM | Synthetic | Three features | 99.26% | Only layer 3 feature collection duplicates dataset instances. |
| [31] | Detection | LSVM & LR | Synthetic | Four features | 98% | No mitigation. |
| [32] | Detection | SVM | Synthetic | Five features | 97.5–99.9% | No mitigation. |
| [33] | Detection | SVM KNN DT NB RF | Synthetic | Six features | 99.84% 98.96% 99.26% 77.64% 99.19% | No mitigation. |
| [34] | Detection/Mitigation | ML | NSL-KDD dataset | 25 features | F1-score (77%) | High complexity due to many features. |
| [35] | Detection | polynomial SVM | Synthetic | Five features | 95.38% | No mitigation. |
| [36] | Detection/Mitigation | SVM- SOM | Synthetic | Not mentioned | 98.12% | Features not defined. |
| [37] | Detection/Mitigation | RF | Synthetic | Five features | 98.38% | Evaluation of only one model. |
| [38] | Detection | SVM DT | KDD99 | 41 features | 78% 85% | High complexity due to many features. No mitigation and accuracy not high. |
| [39] | Detection | SVM | Synthetic | Six Features | 95.24% | Only layer 3 feature collection duplicates dataset instances. |

## 4. Motivation

One of the most important parts of managing an FON is the DCN network. Data communication networks (DCN) are utilized by the network management system (NMS) to monitor, run, and maintain FON operations. Consequently, when a DDoS attack occurs excessively in the DCN, NMS becomes inaccessible and FON becomes invisible and out of control. DDoS attacks are harmful to every network. They can use many ways to attack a specific network, such as ICMP flooding, TCP SYN flooding, and UDP flooding, all of which are damaging to DCN networks. Physical FON equipment can be damaged by DDoS attacks as well.

The present research suggests utilizing Machine Learning to detect a DDoS attack in an FON network using a small group of features. Fewer features enable faster processing of network packets to categorize them as either an attack or regular traffic. This categorization is vital, as it is essential to detect a DDoS attack as soon as is feasible. Our suggested method can detect a DDoS attack with 100 percent accuracy using only five features. Extremely effective distributed denial of service (DDoS) attacks have the potential to crash the target FON server or significantly slow its performance for an extended period of time. In the second quarter of 2022, for example, network-layer DDoS attacks climbed by 109% annually. Furthermore, Quarter-over-Quarter (QoQ) attacks of 100 Gbps or more and attacks lasting more than 3 h grew by 8% and 12%, respectively [40]. Telecommunications, gaming/gambling, and information technology and services were the most targeted industries. Due to the advanced features that SDN provides, such as a global view of the network, software-based management and examination of network traffic, and dynamic updating of forwarding rules, detecting and mitigating DDoS attacks is becoming more feasible.

## 5. Methodology

ML and SDN together can protect an FON from DDoS attacks. Figure 5 shows the steps, starting with setting up the network by assessing the selected models and ending with implementing and evaluating the proposed solution. For each step or phase in this research, appropriate tools and techniques are used.

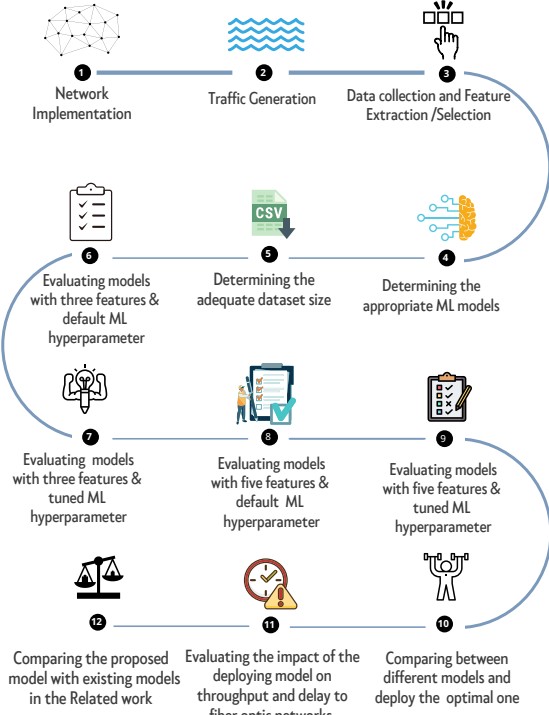

**Figure 5.** Research methodology.

### 5.1. Network Implementation

The suggested topology for this project is an FON with SDH network elements connected to the NOC through a DCN. The main goal of this project is to protect the external DCN, which allows the NOC engineers to control, maintain, and monitor the SDH network through the NMS, from DDoS attacks. Figure 6 illustrates our suggested network topology, which is made up of SDH nodes split into three areas. Each area has several nodes and a single gateway to the outside DCN.

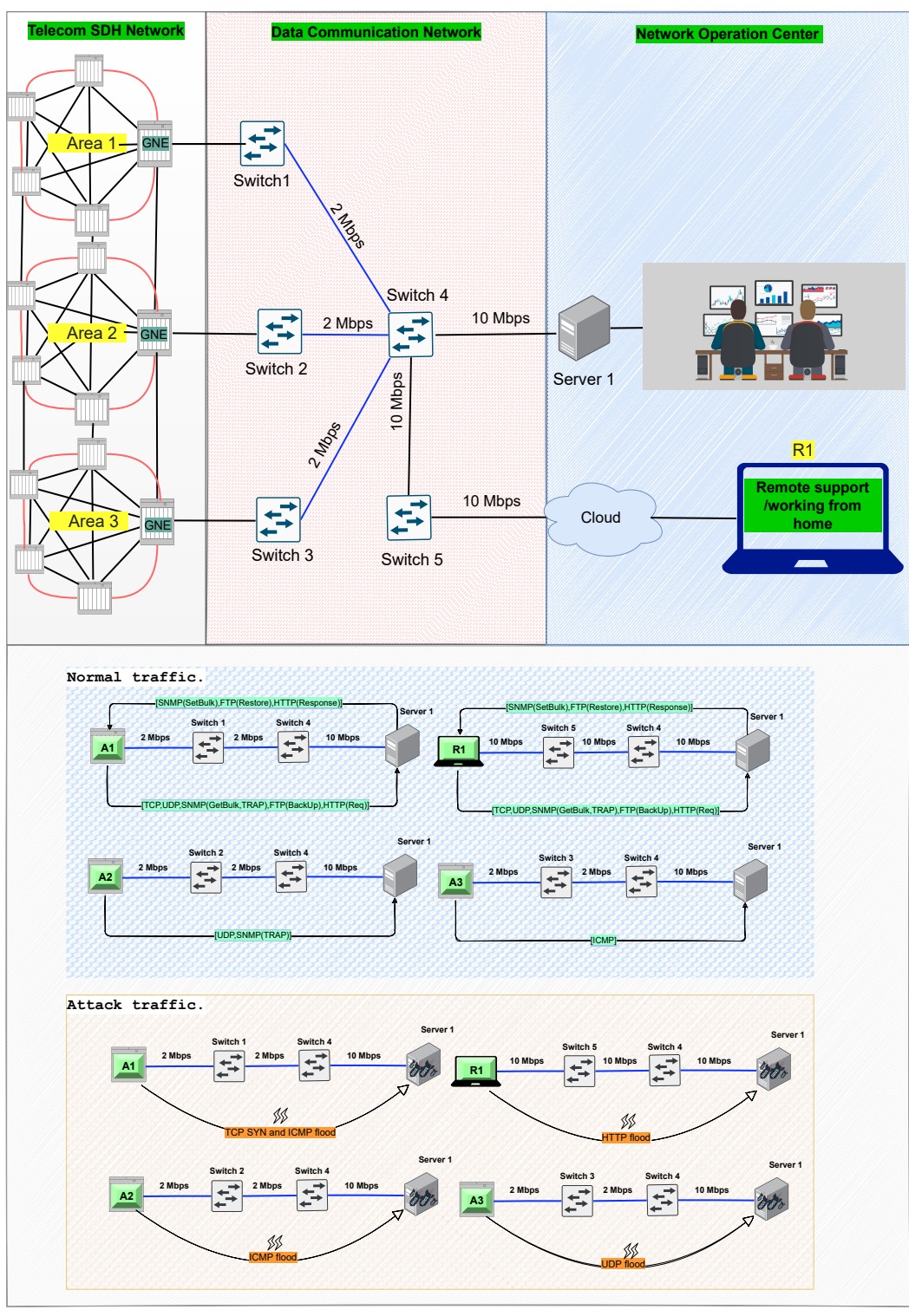

**Figure 6.** Suggested topology and normal/attack traffic scenarios.



The multiplex section DCC works as a single 576 Kbit/s message-based channel using the section overhead bytes D4 through D12 [14]. First, we assume that the link bandwidth to the NOC center is set to 2 Mbps to avoid congestion, although 576 kbps is enough for this experiment. Next, the gateway nodes (GNEs) are connected to an external DCN that leads to a switch at the NOC center. The DCN is connected to the server, which runs the NMS application that allows network operators to configure, manage, and monitor the SDH network. We assume that the server is the victim of a DDoS attack arriving from the SDH network, cloud-based applications (remote support), or switches. Several different DDoS attacks are considered, including HTTP flood from the R1 side, UDP flood from area 2, ICMP flood from area 3, and TCP-SYN and ICMP flood from area 1.

The proposed work was simulated on a Mac laptop with an Intel® CoreTM i75500U processor and 4 GB of RAM. Linux Ubuntu 14.04 was used as the Virtual Machine operating system. Mininet version 2.2.1 was used as the network emulator. Mininet was used as a standard network emulator to build the suggested topology in the SDN environment. An RYU controller was selected to control the flows into the network and act as an SDN controller.

The following tools are used in this step:

- **Python** is a high-level open-source programming language initially developed by Guido van Rossum in 1991. It can be used to support machine learning through a number of libraries and tools. Thanks to its wide range of machine learning programming capabilities [41], Python has developed into a potent programming language with support for object-oriented, imperative, functional, and procedural development methods. In addition, Python now has built-in libraries for different machine learning algorithms.

- **Mininet** is a popular network emulator tool for SDN research, and is used in our research as it makes for an excellent underlying network topology. The mininet environment enables the creation of virtual hosts and switches, which can then be connected to create the desired network topology. This tool has a Python API as well, making it easy to create custom topologies and experiments [31]. As shown in this Listing 1, a mininet network emulator can be used to build a network consisting of virtual hosts, switches, controllers, and connections. OpenFlow is enabled on mininet hosts' switches because the mininet hosts run standard Linux software, enabling SDN and highly flexible custom routing. The primary benefit of using mininet is that it supports Open-Flow Protocol, which SDNs require to configure their networks and perform computations. It provides an affordable way to build, test, and create custom network topologies that closely resemble real networks.

**Listing 1.** Mininet code example.

```
#Example of adding Switch
s1 = self.addSwitch('s1')
#Example of adding Host
h4=self.addHost('h4',ip='10.1.1.4/24',mac="00:00:00:00:00:04")
#Example of adding Link
self.addLink(h1, s1, cls=TCLink, bw=10)
```

- **Ryu controller** is a component-based SDN framework, SDN architecture, and programmable controller tool. It is well-defined application programming interfaces (APIs) for software components that makes it simple for developers to build new apps for network management and control. In addition, Ryu is compatible with a variety of protocols for the management of network devices. These protocols include OF-config, Netconf, and OpenFlow (particularly versions 1.0, 1.2, 1.3, 1.4, and 1.5 and the Nicira extensions). Because it is licensed under Apache 2.0, the Ryu code can be used by anybody without cost [42]. This license makes the network more flexible by making it easier to manage, handle, and adapt traffic flows. Listing 2 shows an example flow.

**Listing 2.** Viewing flows in Ryu Controller.

```
#Example of MAC Flows Match (Layer #2)
cookie=0x0, duration=3.782s, table=0, n_packets=5, n_bytes=136, priority=1,in_port
    ="s1-eth1",dl_src=00:00:00:00:12:13,dl_dst=00:00:00:00:11:21 actions=output:"
    s1-eth2"

#Example of IP Flows Match (Layer #3)
cookie=0x0, duration=15.987s, table=0, n_packets=5, n_bytes=197, priority=1,ip,
    nw_src=192.168.0.1,nw_dst=192.168.0.2 actions=output:"s1-eth1"

#Example of TCP/UDP Flows Match (Layer #4)
cookie=0x0, duration=4.933s, table=0, n_packets=25852, n_bytes=11049192464,
    priority=1,tcp,nw_src=192.168.0.9,nw_dst=192.168.0.1,tp_src=37704,tp_dst=5001
     actions=output:"s1-eth1"

#command to show flow :
--@--:~$ sudo ovs-ofctl -O OpenFlow13 dump-flows s#

#connection info with controller:
cookie=0x0, duration=7.781s, table=0, n_packets=89, n_bytes=4112, priority=0
    actions=CONTROLLER:65535
```

### 5.2. Traffic Generation

In this step, a dataset is created by simulating normal and attack traffic with several traffic generation tools. In Section 2.5, we have already discussed the types of normal and attack traffic in the DCN of an FON. Normal traffic is expected in the FON between the nodes and the NMS (i.e., TCP, UDP, SNMP, FTP, and HTTP). Our experiment desires to prevent all types of DDoS traffic, namely, TCP, UDP, HTTP, and ICMP flood.

The traffic generation procedure is accomplished within the four different scenarios described below, and is shown in Figure 7.

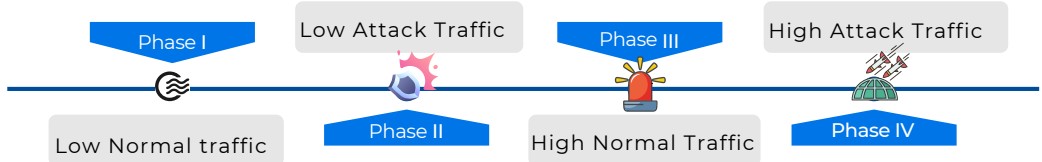

**Figure 7.** Traffic generation phases.

I   **Low normal traffic:** this occurs when there is little traffic between nodes and servers in an FON in an ideal traffic case, which means that there are not very many activities, alarms, or modifications in the configuration of the network. Up to five simultaneous connections from a single region to the server are considered as low normal traffic.

II   **Low attack traffic:** this occurs when there is a small group of attackers attacking a target and the target has enough resources to reply to the attackers. An attack with low traffic, such as an ICMP flood, can slow down a server's performance. A low attack is defined by 25 to 33 spoof packets per second.

III   **High normal traffic:** this occurs when there are more activities on the FON side, such as fiber cuts, traffic needing to be rerouted, or an NOC engineer needing do carry out a monthly network backup. Six to ten simultaneous connections are considered high normal traffic.

IV   **High attack traffic:** when a group of hackers constructs a DDoS attack on a FON, they flood the server with a large amount of traffic, restricting the network operators in their management of the FON. A high attack is estimated as 50 to 100 spoof packets per second.

Listing 3 shows how the traffic types are generated using existing tools; here, the target IP is 10.1.1.4.

**Listing 3.** Traffic generation phases.

```
#PHASE I: Low−normal traffic
iperf −c 10.1.1.4 −t $(shuf −i 1−5 −n 1) # one to five connections at same time.
#PHASE II: Low−attack traffic
hping3 −2 −−rand−source −i u$(shuf −i 30000−40000 −n 1) −d $(shuf −i 60−100 −n 1)
    −c $(shuf −i 1000−1500 −n 1) 10.1.1.4 # 25 to 33 spoofed packets/second
#PHASE III: High−normal traffic
iperf −c 10.1.1.4 −t $(shuf −i 5−10 −n 1) −P 5 # five to ten connections at same
    time.
#PHASE IV: High−attack traffic
hping3 −2 −−rand−source −i u$(shuf −i 10000−20000 −n 1) −d $(shuf −i 60−100 −n 1)
     −c $(shuf −i 1000−1500 −n 1) 10.1.1.4 # 50 to 10 spoofed packets/second
```

The following tools are used in this step:

- **Iperf** is an open-source network performance measurement tool. It sends traffic from one host to another with the adapted bandwidth to obtain the desired results. Iperf can make more measurements than only the throughput, including packet loss, jitter, and how the traffic is distrbuted. Iperf works for TCP and UDP traffic, is currently built into mininet, and has differences for each protocol [43,44].
- **Distributed Internet Traffic Generator (D-ITG)** is a tool that can generate packet-level traffic that can be precisely replicated [43,45]. D-ITG contains models created to mimic the sources of several protocols, including TCP, UDP, ICMP, DNS, Telnet, and FTP. In addition, D-ITG can simulate FTP traffic (e.g., backup and restore) and SNMP traffic (e.g., set bulk and get bulk).
- **SNMP TRAP** is a tool that generates a notification (i.e., a trap) report alongside an event or alarm to the SNMP manager with the specified message [46].
- **Curl** is a command line tool used to effectively simulate HTTP traffic [47].
- **Hping3** is an open-source packet generator and analyzer for the TCP/IP protocol created by Salvatore Sanfilippo. It can be utilized to benefit from an optimal scanning technique, and is even included in the Nmap Security Scanner [48] as one of the standard tools for security auditing and testing of firewalls and networks. Furthermore, as explained in Section 2.5, the Hping3 tool can simulate all DDoS attacks. Listing 4 shows how to use the associated tools to generate normal or attack traffic.

**Listing 4.** Normal and attack traffic generation.

```
#Normal traffic

iperf [−u UDP] [−b bandwidth] [−c target_IP] [−t time] # UDP
iperf [−c target_IP] [−t time] # TCP
snmptrap [ −a host ] [ −h target_IP ] [ −c community ] [−v version ] −m message#
    snmp_trap
ITGSend  [−T Protocol] [−a target_IP] [−Fp filename] [−rp target_port} #ftp_backup
    /restore snmp GetBulk/SetBulk
hping3 [target_IP] [−− Protocol] [−c packet count] [−d data size packet body size]
    [−s baseport] [−k still keep source port] [−p target_port]    normal ICMP
    [−1] or UDP [−2] or TCP []
Curl [target_IP or URL]    # HTTP

#Attack traffic

hping3 [target_IP] [−− Protocol] [−c packet count] [−d data size packet body size]
    [−s baseport] [−k still   keep source port [−p target_port] #TCP SYN & ICMP
    Flood attack
hping3 −−rand−source [−S syn] [−i interval wait (uX for X microseconds] [−d data
    size packet body size [−c packet count] [target_IP] #UDP flood attack
hping3 −−rand−source [−2 udp] [−S syn] [−i interval wait (uX for X microseconds]
    [−d data size packet body size] [−c packet count] [target_IP] #ICMP flood
    attack
hping3 −−rand−source [−1 ICMP] [−S syn] [−i interval wait (uX for X microseconds]
    [−d data size packet body size] [−c packet count] [target_IP] #ICMP flood
    attack
```

```
hping3 --rand-source [-p 80] [-S syn] [-i interval wait (uX for X microseconds] [-
    d data size packet body size [-c packet count] [target_IP] #HTTP flood attack
```

### 5.3. Data Collection and Feature Extraction/Selection

This step is actually the most critical stage of the design, as it impacts the results of all subsequent steps. Feature extraction within this phase is responsible for calculating the feature values from the switch flow table and creating a matrix containing these values. In the SDN environment, an OpenFlow protocol assists in creating a flow table that contains status information collection. The switch responds to the SDN controller and periodically sends request messages to obtain flow statistics. The time interval between receiving the flow tables should be moderated in order to collect the status information of the flow table. To achieve this, the "Sudoovs-ofct1 dump-flows s1" command is run on the SDN RYU controller. An example of the extracted information is shown in Listing 2.

In the SDN controller, an incoming network traffic flow is identified along with a number of features. These features can be extracted and collected during the traffic generation phases. These preselected features became essential for distinguishing between normal and attack traffic, as mentioned in prior studies [26,27,30,32,39]. These features, provided below, were monitored and collected for each 5 seconds during traffic generation phases:

1.  **Speed of the source IP (SSIP)** can be determined by the number of IP addresses received from sources in a certain amount of time; it can be defined as Equation (1) [27,30,32,39]:

$$SSIP = \frac{\sum IPsrc}{T} \tag{1}$$

    where $T$ represents the time between samples and $IPsrc$ is the total number of IP sources incoming in the received flows. DDoS attacks initiate many distributions of data packets duplicated randomly, leading to many attacks and an immediate increase in the number of source IP addresses.

2.  **Speed of session (SOS):** the number of flow entries in the transport layer (L4 in OSI) determines how many sessions are opened per unit of time $T$. This feature, introduced in our research and named SOS, is defined as shown in Equation (2):

$$SOS = \frac{\sum L4\_Sessions}{T} \tag{2}$$

    This feature is essential for identifying attacks, as the number of open sessions per unit of time $T$ may considerably increase during a DDoS attack.

3.  **Ratio of pair-flow entries (RPF)** is the total number of interactive flow entries (i.e., bi-directional) divided by the total number of IP addresses during the period $T$; it is defined by Equation (3) [27,32]:

$$RPF = \frac{intIPs}{N} \tag{3}$$

    Under normal conditions, the traffic between sources and destinations is usually interactive. Therefore, bidirectional flows induce a number of flows that is equal to the number of IP addresses in the network. However, when an attack occurs, interactive communication is disabled and unidirectional flows (from multiple sources to a destination) become established in the network. In this case, the number of IPs is much larger than the number of interactive flows. In this manner, due to the low number of interactive flows and no service availability, the traffic can be characterized as a DDoS attack.

4.　**Standard deviation of flow of packets (SDFP)** during a period T is defined by Equation (4) [27,32,39]:

$$SDFP = \sqrt{\frac{1}{N}\sum_{i=1}^{n}(packet_i - Mean\_packets)^2} \tag{4}$$

where $packet_i$ is the total number of packets of the $i$th flow and $Mean\_Packets$ is the average number of packets in the network within period $T$. Due to the substantial correlation between this feature and an attack, the standard deviation is lower for an attack compared to normal traffic.

5.　**Standard deviation of flow bytes (SDFB)** is calculated by the difference between the number of bytes in a flow compared to the average number of bytes per flow during a period $T$. It is defined by Equation (5) [27,32,39]:

$$SDFB = \sqrt{\frac{1}{N}\sum_{i=1}^{n}(bytes_i - Mean\_bytes)^2} \tag{5}$$

where $Bytes_i$ is the number of bytes in the $i$th flow and $Mean\_Bytes$ is the average number of bytes during a period $T$ in the network.

Both the standard deviation of the packets in flows and the standard deviation of bytes in flows significantly affect DDoS attack. However, the predictive value of SDFP is significantly lower during attacks than during normal traffic.

### 5.3.1. The Dataset

Our dataset was constructed using the four different scenarios outlined in Section 5.2: low normal traffic, low attack traffic, high normal traffic, and high attack traffic. In addition, in Section 2.5 we have described the traffic types in fiber optic networks (SNMP, FTP, HTTP, TCP, and UDP as Normal traffic and TCP-SYN, UDP, TCP, and HTTP floods as attack traffic). Figure 8 shows the various traffic types during the traffic generation phases.

As illustrated in Figure 8, there are no sudden increases of packets over time in normal traffic. However, there are always rapid rises during attack traffic, even up to the maximum network bandwidth. In this case, a number of packets are discarded and retransmitted by the congestion control mechanisms. The increases in these retransmissions further increases the network load.

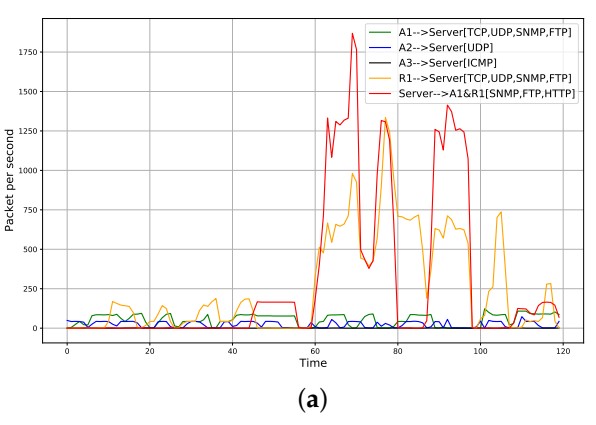

(**a**)

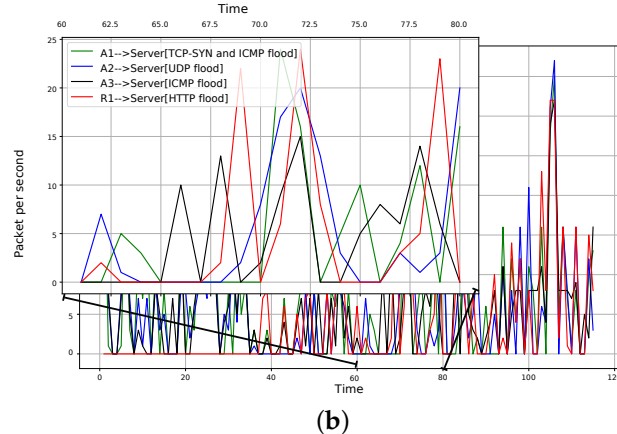

(**b**)

**Figure 8.** *Cont.*

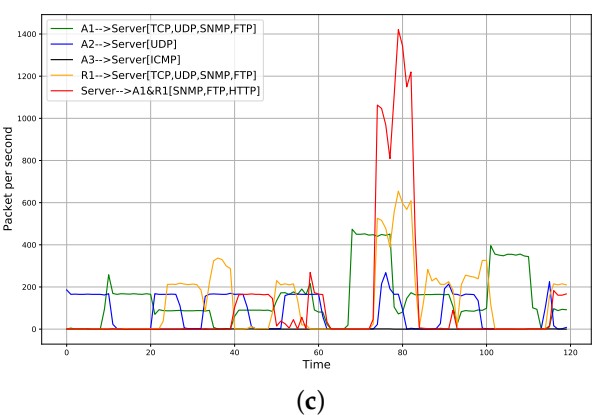

(**c**)

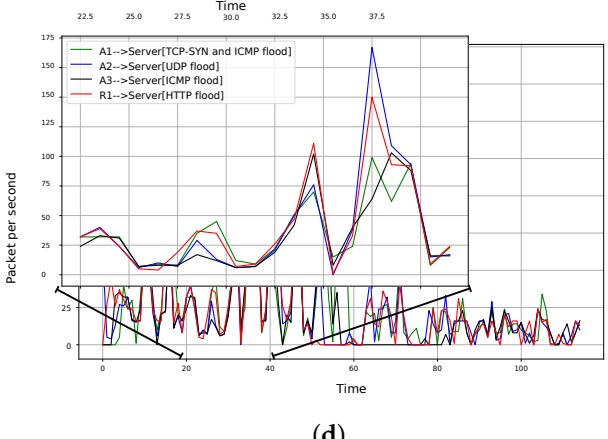

(**d**)

**Figure 8.** Traffic generated during the four phases: (**a**) low normal traffic; (**b**) low attack traffic; (**c**) high normal traffic; (**d**) high attack traffic.

As shown in Figure 8, the time of traffic generation for each phase is fixed to 120 s. This procedure is repeated four times. The selected features are then collected from all switches every five seconds by an SDN controller. Finally, the features are published in a CSV file. This procedure generates 480 instances, produced as follows: (1) 120 s are sampled into separated 5 s time intervals, resulting into 24 instances; (2) features are collected from five switches, resulting in 120 instances; and (3) the same procedure is repeated four times, resulting into 480 instances. Then, the dataset is manually cleaned to remove redundant samples; this improves model performance, as predictions are based on unseen data. This leaves 400 instances for each step and a total of 1600 instances, as illustrated in Figure 9.

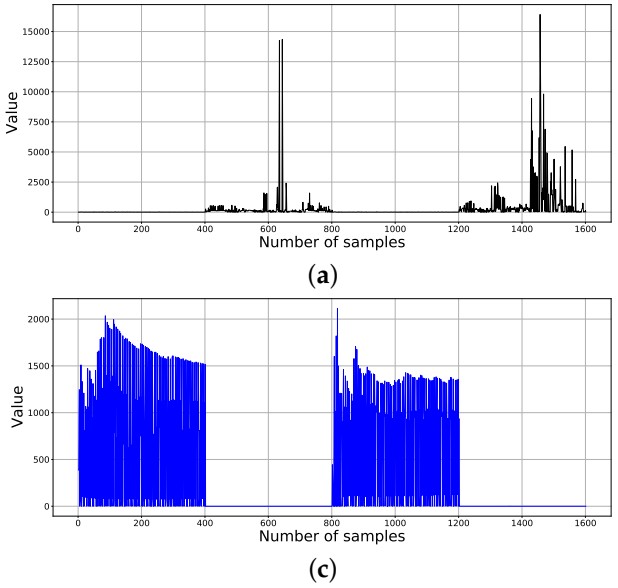

(**a**)

(**c**)

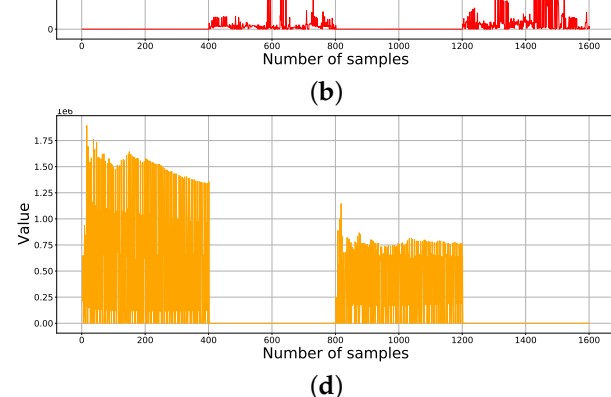

(**b**)

(**d**)

**Figure 9.** *Cont.*

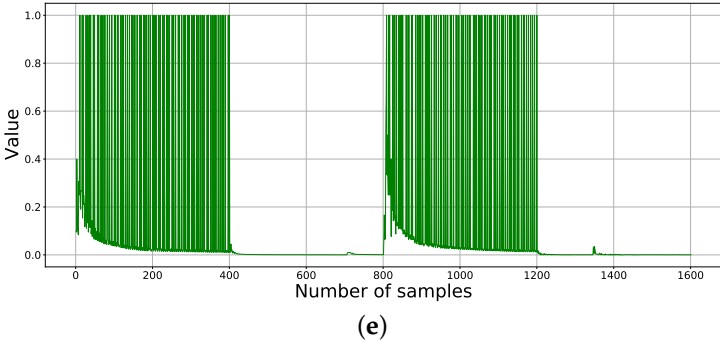

(**e**)

**Figure 9.** Feature values under normal and attack traffic: (**a**) speed of sessions; (**b**) speed of source IP; (**c**) standard deviation of flow packets; (**d**) standard deviation of flow bytes; (**e**) ratio of pair-flow entries.

### 5.3.2. Selected Features

The selected features are described in Section 5.3, with a novel feature known as SOS introduced and calculated based on the number of flow entries at the transport layer. Five features were collected during the traffic generation phase, and 1600 instances of these features were gathered. The resulting dataset consists of five columns: SOS, SSIP, RPF, SDFP, and SDFB. An example of normal and DDoS traffic features is provided in Table 2. In the type column, the label '0' represents normal traffic and the label '1' represents DDoS attack traffic.

**Table 2.** Samples of the dataset.

| SOS | SSIP | RPF | SDFP | SDFB | TYPE | PHASE |
|-----|------|-----|------|------|------|-------|
| 7 | 1 | 0.083333 | 78.5163 | 11,8931.2 | **0** | |
| 19 | 0 | 0.307692 | 1007.227 | 670,561.4 | **0** | |
| 8 | 1 | 0.25 | 1510.026 | 939,150 | **0** | Low Normal Traffic (I) |
| 12 | 1 | 0.333333 | 99.04024 | 150,217.5 | **0** | |
| . | . | . | . | . | . | |
| . | . | . | . | . | . | |
| 107 | 107 | 0.002331 | 0.048252 | 4.728723 | **1** | |
| 123 | 123 | 0.002513 | 0 | 0 | **1** | |
| 113 | 113 | 0.002255 | 0.047458 | 4.650868 | **1** | Low Attack Traffic (II) |
| 111 | 111 | 0.00216 | 0.046449 | 4.551984 | **1** | |
| . | . | . | . | . | . | |
| . | . | . | . | . | . | |
| 12 | 4 | 0.166666667 | 268.1955528 | 149,167.4674 | **0** | |
| 7 | 2 | 0.064516129 | 23.52020408 | 35,562.54857 | **0** | |
| 5 | 2 | 0.066666667 | 0.577350269 | 567.0716445 | **0** | High Normal Traffic (III) |
| 10 | 1 | 0.25 | 93.32372 | 141,228.5 | **0** | |
| . | . | . | . | . | . | |
| . | . | . | . | . | . | |
| 196 | 211 | 0.006494 | 0.113402 | 13.72169 | **1** | |
| 182 | 197 | 0.00409 | 0.078165 | 10.47416 | **1** | |
| 20 | 19 | 0 | 0.917663 | 109.2019 | **1** | High Attack Traffic (IV) |
| 236 | 251 | 0.008287 | 0.074175 | 9.444982 | **1** | |
| . | . | . | . | . | . | |
| . | . | . | . | . | . | |

Under normal circumstances, the speed of sessions (SOS) is not significant. In addition, the ratio of iterative connections should be around one, as it represents a two-way connection, which is considered real traffic. However, random spoofing sessions and new IPs are

created during DDoS attacks, resulting in an increased number of SSIPs and SOS. On the other hand, there is a strong link between SDFB and DDoS attacks. The attacker sends many small attack data packets, meaning that the standard deviation is smaller than with normal data packets. Additionally, if a DDoS attack happens, the amount of data coming into the server at time T increases quickly, and the server becomes unable to perform the requested services. Therefore, there is a sudden drop in the number of interactive flows. The features of the dataset are shown in Figure 9. The generated dataset contains 1600 random data points. Samples 1–400 indicate low normal traffic, samples 401–400 represent low attack traffic, samples 801–1200 represent high normal traffic, and samples 1201–2600 represent heavy attack traffic. During the traffic generation phases, these features were captured every 5 s.

When preparing data for supervised machine learning algorithms, dataset partitioning is a crucial first step. The normal practice is to choose between 20% and 30% of the data for testing and the rest for training. In this study, the "train_test_split" function in scikit-learn tool was used with a fixed "random_state" to ensure that all models used the same training and testing subsets.

### 5.3.3. Comparative Study of the Most Popular ML Models

Considering the available machine learning algorithms, no one solution or approach clearly fits our presented problem. For example, algorithm selection requires checking the size of dataset, whether the data are labeled or unlabeled, and whether the problem type is a classification or regression problem.

The size of our dataset is 1600 instances, with five features and two classes. It was only after answering these questions that the classification algorithms could be decided upon. As a result, six existing popular ML algorithms which are frequently used to solve straightforward classification problems were selected, as described below:

- Logistic Regression (LR) is a well-known machine learning model used for binary classification. It has a probabilistic framework able to adapt classification thresholds and obtain conviction intervals [31].
- K-Nearest Neighbors (KNN) is a non-parametric supervised learning model used to solve classification and regression problems. It performs classification and prediction of new datapoints by comparing them to predefined groups based on the proximity principle [33].
- Support Vector Machine (SVM) works by finding the best way to separate labeled instances in each dataset using hyperplanes [35]. Various kernel functions, including linear, polynomial, and radial-based, can be used in the mapping process [3].
- Naive Bayes (NB) is a typical classifier founded on Bayes' theorem. It is able to define the probability of an event based on previous facts. Naive Bayes classifiers rely on the hypothesis that the features are independent [49,50].
- Decision Tree (DT) is a method of classification that uses a tree structure. This algorithm divides the population into two or more groups based on the most critical attributes [51].
- Random Forest (RF) is a regression-based machine learning algorithm guided by a base model series. Random forest regression has many benefits, such as high accuracy, efficiency, and performance when dealing with essential variables. To obtain the best prediction results, it is vital to choose the correct number of trees [52].

### 5.3.4. Determining the Adequate Dataset Size

The size of the dataset is one of the most significant problems in supervised learning. Obviously, insufficient training data can lead to a bad approximation, negatively impacting ML model performance. Another training factor is the time it takes to train the model. The time interval for training an ML model depends on the size of the database (i.e., it increases with the number of samples in the dataset). Thus, training the model in a faster manner might consider fewer data, while more data may provide better results.

Learning curve graphs illustrate how well a model learns over time or with more experience. Learning curve graphs are often used as diagnostic tools to investigate how the performance of a model changes with dataset size. Figure 10 shows graphs of the learning curves of selected machine learning algorithms used in this research, demonstrating that a small dataset size is acceptable for the DT and RF algorithms. However, the other algorithms need a more extensive dataset size. When using more than 1000 samples for training sets, all selected ML algorithms show almost the same accuracy level. Therefore, this number of samples was considered sufficient for performance evaluation, and was used during the next step for model improvement.

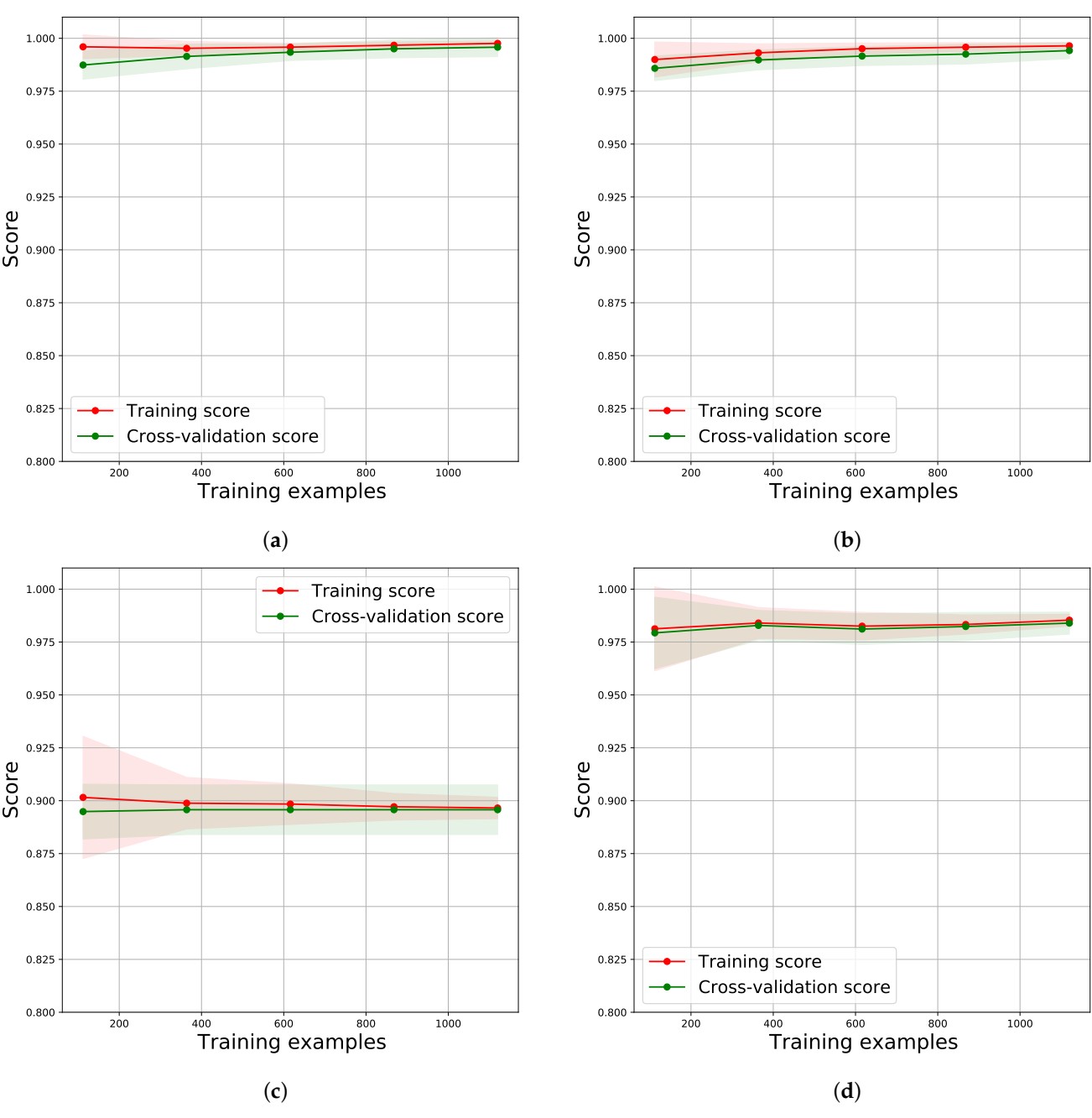

**Figure 10.** *Cont.*

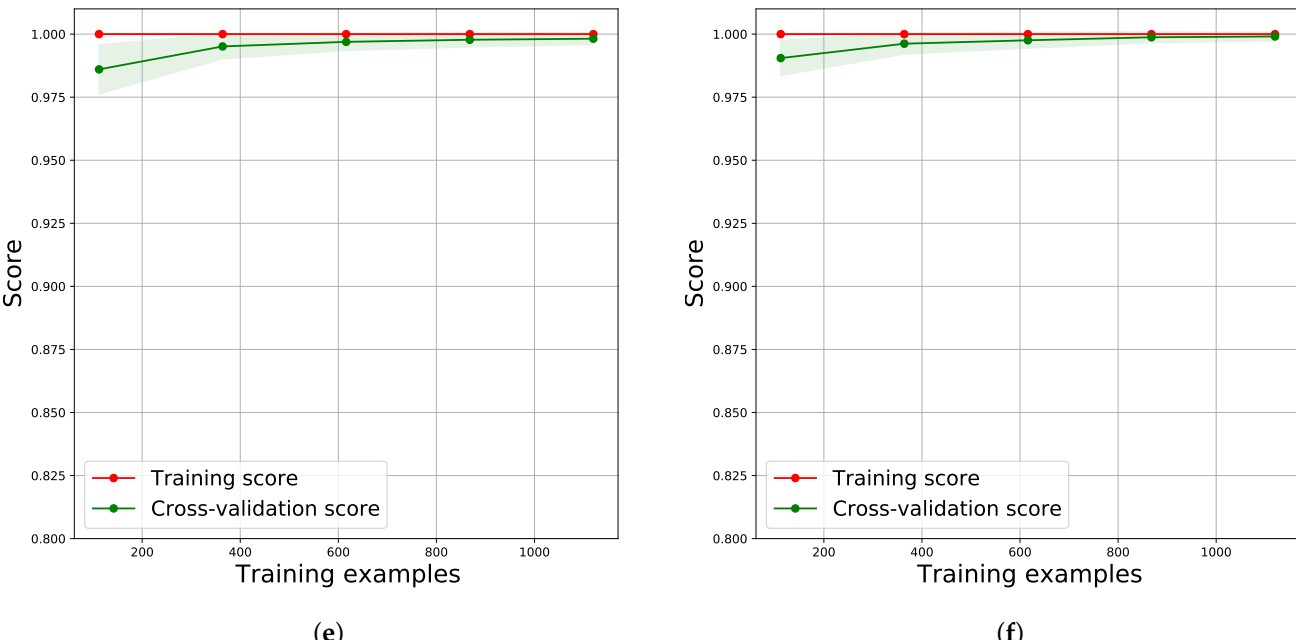

(**e**)                  (**f**)

**Figure 10.** Learning curves of selected algorithms: (**a**) learning curves_LR; (**b**) learning curves_KNN; (**c**) learning curves_SVM; (**d**) learning curves_NB; (**e**) learning curves_DT; (**f**) learning curves_RF.

### 5.4. Performance Metrics

Several metrics were considered for performance evaluation during the training and testing phases: accuracy, precision, recall, and F1 score. A confusion matrix was used to determine how to measure these metrics. A confusion matrix, sometimes known as an error matrix, is a table summarizing the results of an algorithm when processing data samples [53]. Additional metrics, such as learning curves, Cross-validation scores, and the standard deviation of cross-validation scores, can be employed to assist in the initial selection of dataset size and ML models. Usually, a supervised learning model is represented based on how the machine learning model responds. This matrix has different cases, such as True Positive, True Negative, False Positive, and False Negative. These cases are described as follows:

- **True Positive (TP):** the number of attacks that the classifier predicts correctly.
- **True Negative (TN):** the amount of normal traffic that the classifier predicts correctly.
- **False Positive (FP):** how often normal traffic is mistakenly labeled as attack traffic.
- **False Negative (FN):** how often attack traffic is mistakenly labeled as normal traffic.

The main measurements used to evaluate how well the learning model works are:

1. **Learning curve:** a learning curve shows how an estimator's validation score and training score change as the number of training samples changes (i.e., with dataset size). This curve is calculated from the training data in order to inform how well a model is learning as the amount of data increases. Moreover, it determines whether the estimator is more likely to make a bias or variance error.

2. **Accuracy:** the ratio of the number of correct predictions to the total number of predictions. It can be calculated by Equation (6):

$$Accuracy = \frac{TP + TN}{TP + FP + TN + FN} \tag{6}$$

3. **Precision:** the number of true positive cases out of all predicted positive cases. The precision value, which is between 0 and 1, can be calculated using Equation (7):

$$precision = \frac{TP}{TP + FP} \tag{7}$$

4. **Recall:** the number of predicted positive cases as a percentage of all positive cases; it is similar to the True Positive Rate (TPR), and is calculated using Equation (8):

$$Recall = \frac{TP}{TP + FN} \tag{8}$$

5. **F1-score:** the F1-score represents the harmonic mean of precision and recall. It accounts for both false positives and false negatives. Consequently, it performs effectively on uneven datasets. It can be determined by Equation (9):

$$F1 = \frac{2 \times (precision \times recall)}{(precision + recall)} \tag{9}$$

6. **Cross-validation score:** a method for re-sampling that uses different parts of the data to test and train a model repeatedly. It is mostly used when the goal is to make a prediction and determine how well a prediction model might work in real-life problems [54].

7. **STD of cross-validation score:** the standard deviation of the cross-validation score measures the variation of the scores when computing a single score for one of the k folds. A low value of this parameter is the most acceptable, and indicates that a dataset is adequate for testing use.

## 6. Results and Discussion

This section analyzes the proposed approach and demonstrates the improvements it makes based on the ML models described in Section 5.3.3. Figure 5 outlines the most important steps followed in our research. The conclusion of this step determines whether it can improve network availability. In the first step, network implementation and traffic generation are realized, followed by a discussion of the desired dataset and evaluation of the models using various features and hyperparameters. The second step consists of improving the proposed approach for better performance.

Table 3 shows the common parameters used for the selected algorithms.

**Table 3.** Common parameters of selected algorithms.

| Machine Learning Algorithm | Parameter [1] | Default Value | Increasing Value | Decreasing Value | Description |
|---|---|---|---|---|---|
| Logistic Regression | tol | float, default = $1 \times 10^{-4}$ | $T$ [2] $:\downarrow A$ [3] $:\downarrow \Rightarrow U$ [4] | $T :\uparrow A :\uparrow \Rightarrow O$ [5] | Tolerance for stopping criteria. |
| | C | float, default = 1.0 | $A :\uparrow \Rightarrow O$ | $A :\downarrow \Rightarrow U$ | Inverse of regularization strength; must be a positive float. As in support vector machines, smaller values specify stronger regularization. |
| K-Nearest Neighbour | n_neighbors | int, default = 5 | $T :\downarrow A :\downarrow \Rightarrow U$ | $T :\uparrow A :\uparrow \Rightarrow O$ | Number of neighbors to use found by Algorithm |
| | p | int, default = 2 | | | Power parameter for the Minkowski metric. This is equivalent to using manhattan_distance (l1) when p = 1 and euclidean_distance (l2) when p = 2. |
| Support Vector Machine | C | float, default = 1.0 | $A :\uparrow \Rightarrow O$ | $A :\downarrow \Rightarrow U$ | Regularization parameter. The regularization strength is inversely proportional to C and must be strictly positive. |
| | kernel | default = 'rb' | | | Specifies the kernel type to be used in the algorithm. |
| | gamma | default = 'scale' | $A :\uparrow \Rightarrow O$ | $A :\downarrow \Rightarrow U$ | Kernel coefficient for 'rbf', 'poly', and 'sigmoid'. |
| Naive Bayes | var_smoothing | float, default = $1 \times 10^{-9}$ | $T :\downarrow A :\downarrow \Rightarrow U$ | $T :\uparrow A :\uparrow \Rightarrow O$ | Portion of the largest variance of all features that are added to variances for calculation stability. |
| Decision Trees | max_depth | int, default = None | $T :\uparrow A :\uparrow \Rightarrow O$ | $T :\downarrow A :\downarrow \Rightarrow U$ | The maximum depth of the tree. If none, then nodes are expanded until all leaves are pure or until all leaves contain less than min_samples_split samples. |
| RandomForest | n_estimators | int, default = 100 | $T :\uparrow A :\uparrow \Rightarrow O$ | $T :\downarrow A :\downarrow \Rightarrow U$ | The number of trees in the forest. |

[1] All parameters can be found in scikit-learn [55]. [2] T: Training time. [3] A: Accuracy. [4] U: Underfiting. [5] O: Overfiting.

### 6.1. Features and Hyperparameter-Based Performance Evaluation

In ML, the smallest subset of components can have a significant effect. Reducing a model's complexity does not always make predictions more accurate. This experiment involves five features and discusses how a different number of features affects model performance. In addition, using the scikit-learn library [55] and grid search mechanism, we evaluate our proposed model with the default hyperparameters and tuned parameters. In this section, we measure the performance metrics of the proposed ML algorithms described

in Section 5.3.3 in four stages (6–9), as shown in Figure 5. In addition, the performance metrics of the selected ML algorithm are shown in Table 4. The main findings regarding ML algorithms are summarized below:

- **LR:** Figure 11a and Table 4 show that performance metrics improved during the four stages (6–9) for the LR algorithm, from 94.59% accuracy at the beginning to a 99.79% accuracy at the end. Adding more features and tuning the LR algorithm's hyperparameters significantly affects the algorithm's performance. However, an increased number of features increases the training time.
- **KNN:** Most of the performance metrics for the KNN algorithm do not become better during the first three stages (6–8). However, in the ninth stage the accuracy, precision, and F1-score all improve, while the cross-validation score and its standard deviation are not enhanced. The training time for KNN is acceptable, as shown in Figure 11b and Table 4.
- **SVM:** As shown in Figure 12a and Table 4, increasing the number of features does not represent adequate input for SVM utilization. When tuning the hyperparameters with "linear" and C equal to "1000", SVM is able to provide high performance. However, training time is a major issue of SVM, especially for detecting DDoS attacks.
- **NB:** The NB algorithm works similarly to LR algorithm. All performance metrics except for precision are excellent in the sixth stage. Figure 12b and Table 4 show the performance in each step. Specifically, the training process of the NB algorithm is very fast compared to the other algorithms.
- **DT:** There is no need to tune any ML hyperparameters, as the performance reached 100% in the sixth and eighth stages. Therefore, this algorithm only has two stages. Figure 13a and Table 4 show that this algorithm works well for protecting FON from DDoS attacks, and the training time is less than most of the selected algorithms.
- **RF:** Similar to the DT algorithm, the RF algorithm works well without changing any ML hyperparameters. A random forest classifier is a robust classifier by default; however, the main problem in this research resides in the numbers and classes being different, meaning that reaching 100% is a reasonable goal. As an advanced step, we changed the default parameters of the algorithm for "n_estimators" from 100 as the default value to 3 (Table 3). This step was performed in order to check the performance with fewer numbers than the default value in the Scikit-learn library. The RF algorithm can be considered as multiple DT algorithms running simultaneously, which is why the training time is higher than the DT algorithm. The performance metrics of this algorithm are shown in Figure 13b and Table 4.

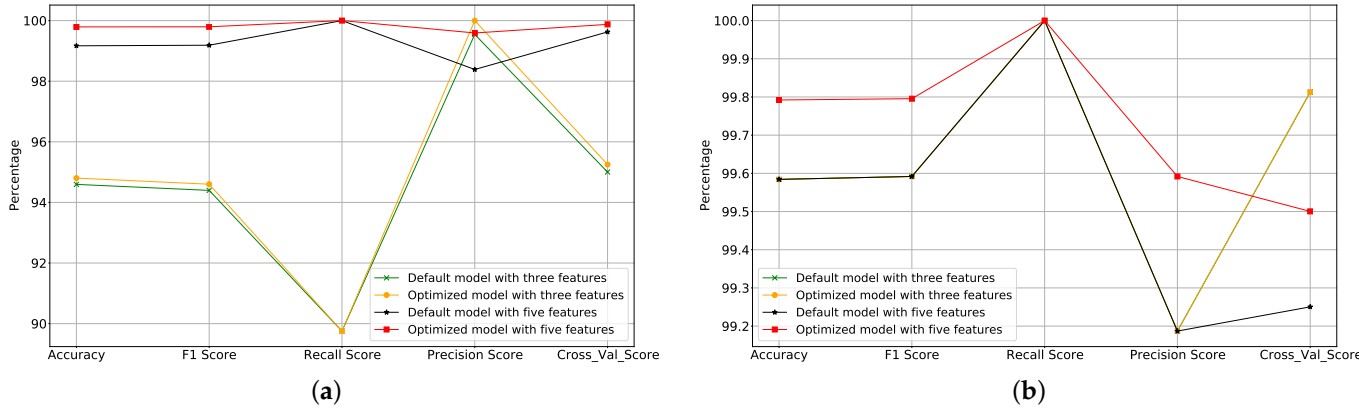

**Figure 11.** Performance evaluation of LR and KNN algorithms: (**a**) LR algorithm; (**b**) KNN algorithm.

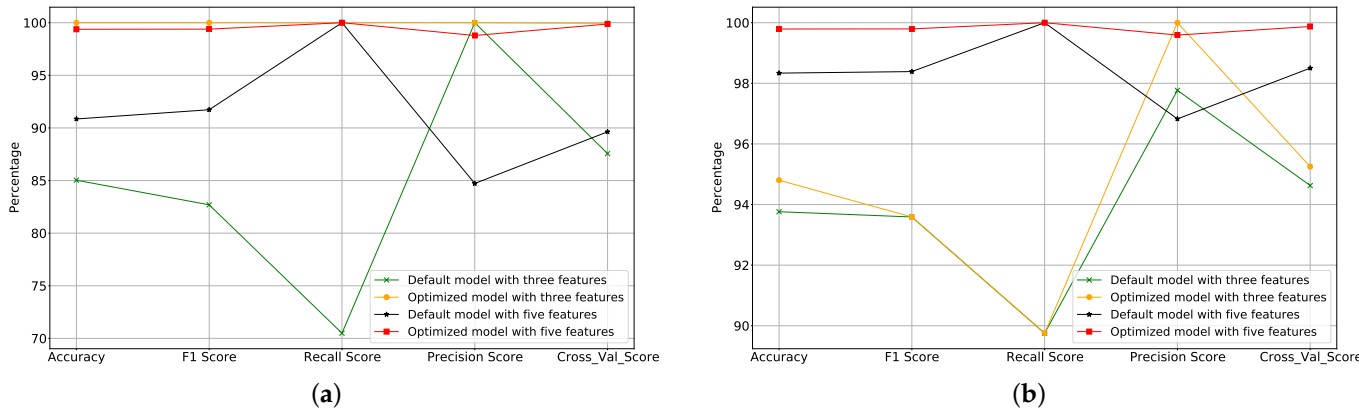

**Figure 12.** Performance evaluation of SVM and NB algorithms: (**a**) SVM algorithm; (**b**) NB algorithm.

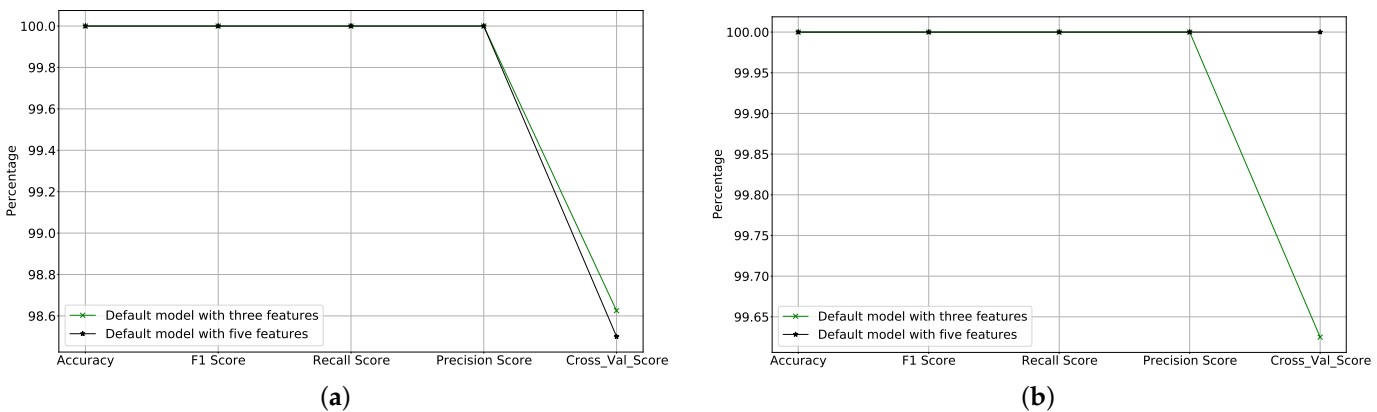

**Figure 13.** Performance evaluation of RF and DT algorithms: (**a**) DT algorithm; (**b**) RF algorithm.

**Table 4.** Performance metrics of selected algorithms.

| MLA | Stage | Numbrt of Features | Grid Searched Hyper-Parameter | Selected Hyper-Parameters | Accuracy | Precision | Recall | F1-Score | CV_Score | Std_CV | Training Time (s) |
|---|---|---|---|---|---|---|---|---|---|---|---|
| LR | 6 | 3 [1] | ['tol': [$1 \times 10^{-2}$, $1 \times 10^{-3}$, $1 \times 10^{-4}$, $1 \times 10^{-6}$]], | Default [3] | 94.59 | 99.55 | 89.75 | 94.4 | 95 | 3.16 | 0.03459 |
| | 7 | 3 | ['C': [0.1,1.0,10.0,100.0]] | O [4] ['C': 100.0, 'tol': 0.01] | 94.8 | 100 | 89.75 | 94.6 | 95.25 | 3 | 0.03057 |
| | 8 | 5 [2] | | Default | 99.16 | 98.39 | 100 | 99.19 | 99.63 | 0.94 | 0.0775 |
| | 9 | 5 | | O ['C': 10.0, 'tol': 0.01] | 99.79 | 99.59 | 100 | 99.8 | 99.88 | 0.25 | 0.0473 |
| KNN | 6 | 3 | ['n_neighbors': [3,5,10,15,20]], | Default | 99.58 | 99.19 | 100 | 99.59 | 99.81 | 0.29 | 0.0033 |
| | 7 | 3 | ['p': [1,2]] | O ['n_neighbors': 5, 'p': 2] | 99.58 | 99.19 | 100 | 99.59 | 99.81 | 0.29 | 0.0010 |
| | 8 | 5 | | Default | 99.58 | 99.19 | 100 | 99.59 | 99.25 | 1 | 0.0020 |
| | 9 | 5 | | O ['n_neighbors': 3, 'p': 1] | 99.79 | 99.59 | 100 | 99.8 | 99.5 | 0.78 | 0.0011 |
| SVM | 6 | 3 | ['C': [0.01,1, 10,100,1000],'kernel': ['linear'], | Default | 85.03 | 100 | 70.49 | 82.69 | 87.56 | 7.28 | 0.0375 |
| | 7 | 3 | 'gamma']: [0.5,0.3,0.2,0.1,0.01], 'kernel': ['rbf'] | O ['C': 1000, 'kernel': 'linear'] | 100 | 100 | 100 | 100 | 99.94 | 0.18 | 14.299 |
| | 8 | 5 | | Default | 90.85 | 84.72 | 100 | 91.73 | 89.63 | 0.93 | 0.0201 |
| | 9 | 5 | | O ['C': 1, 'kernel': 'linear'] | 99.38 | 98.79 | 100 | 99.39 | 99.88 | 0.24 | 1.2329 |
| NB | 6 | 3 | ['var_smoothing': [$1 \times 10^{-2}$, $1 \times 10^{-3}$, | Default | 93.76 | 97.77 | 89.75 | 93.59 | 94.63 | 3.36 | 0.0035 |
| | 7 | 3 | $1 \times 10^{-7}$, $1 \times 10^{-9}$, $1 \times 10^{-11}$, $1 \times 10^{-15}$]] | O ['var_smoothing': $1 \times 10^{-11}$] | 94.8 | 100 | 89.75 | 94.6 | 95.25 | 3 | 0.0039 |
| | 8 | 5 | | Default | 98.34 | 96.83 | 100 | 98.39 | 98.5 | 1.26 | 0.0041 |
| | 9 | 5 | | O ['var_smoothing': $1 \times 10^{-15}$] | 99.79 | 99.59 | 100 | 99.8 | 99.88 | 0.25 | 0.0010 |
| DT | 6 | 3 | | Default | 100 | 100 | 100 | 100 | 98.62 | 3.92 | 0.0036 |
| | 8 | 5 | | Default | 100 | 100 | 100 | 100 | 98.5 | 3.89 | 0.0059 |
| RF | 6 | 3 | | Default | 100 | 100 | 100 | 100 | 99.62 | 0.75 | 0.1699 |
| | – [5] | 3 | | Downgraded [5] [ n_estimators = 3] | 99.79 | 99.59 | 100 | 99.79 | 99.43 | 0.85 | 0.0149 |
| | 8 | 5 | | Default | 100 | 100 | 100 | 100 | 100 | 0 | 0.1800 |

[1] Three features [SOS, SSIP , RPF]. [2] Five features [SOS, SSIP, RPF, SDFP, SDFB]. [3] The default hyperparameter in scikit-learn [55]. [4] The optimized hyperparameters. [5] Downgraded hyperparameter.

### 6.2. Comparison Between Different Models and Deploying the Optimal One

Based on the previous step, the adequate dataset size was determined and the selected ML models were evaluated with the most promising hyperparameters. This step assisted in selecting and deploying the suitable model for the problem raised in this research.

Based on the results of the previous steps, shown in Figure 14 and Table 4, we can have more than one solution to this problem. For example, the training time of SVM with three features (SSIP, SOS, and RPF) and using C = 1000 and a linear kernel was 14.299 s,

while the accuracy reached 100%. On the other hand, when using RF with five features ((SSIP, SOS, SDFP, SDFB, and RPF) and default parameters the training time was 0.18 s and the accuracy reached 100%. Therefore, it can be concluded that the Random Forest algorithm provides the best results in terms of complexity (affordable features), timing, and classification accuracy, especially compared with SVM. Therefore, we proceeded to the implementation phase with the RF algorithm.

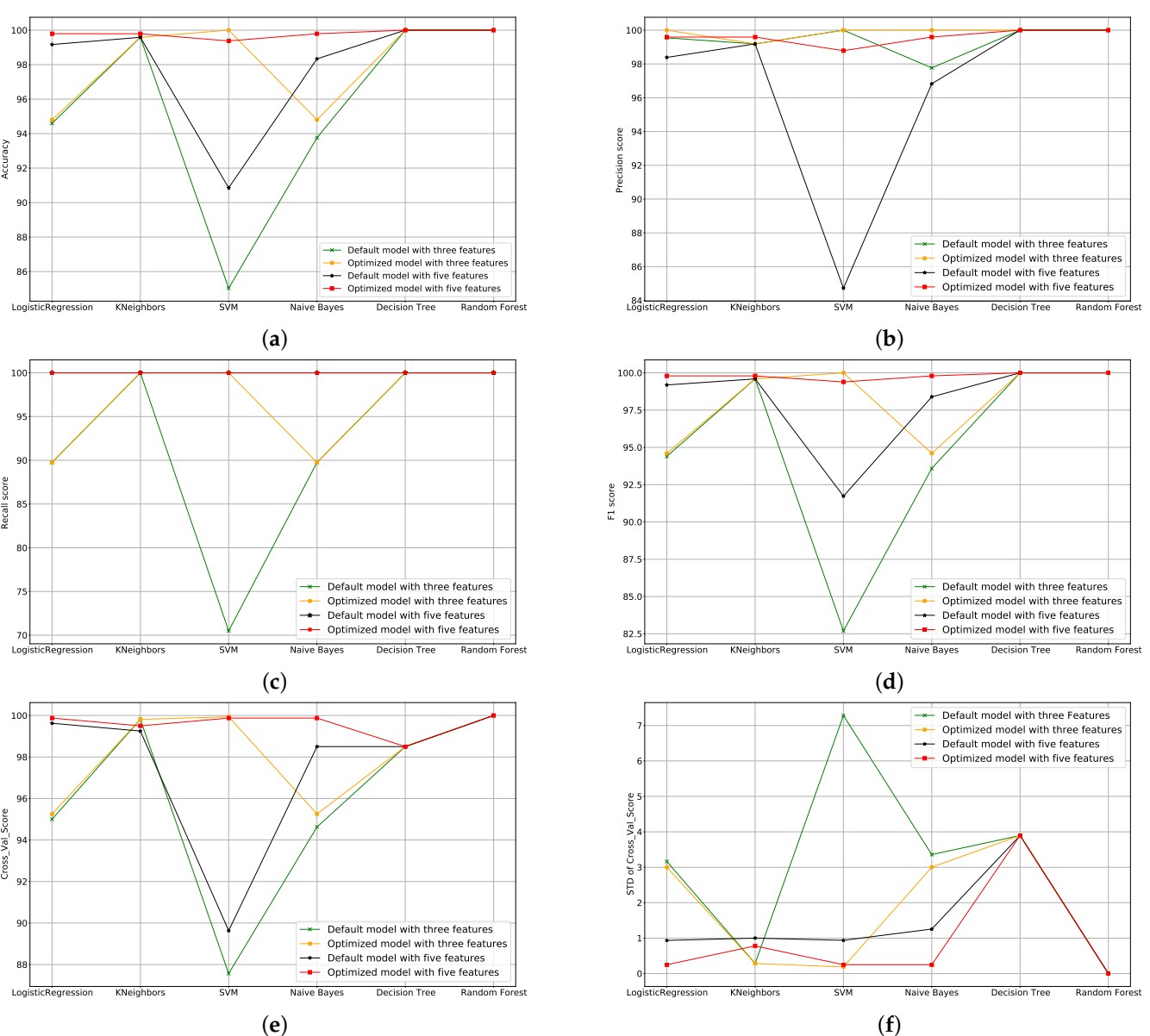

**Figure 14.** Comparison between selected ML models: (**a**) accuracy; (**b**) precision; (**c**) recall; (**d**) F1 score; (**e**) cross-validation score; (**f**) STD of cross-validation score.

### 6.3. Detection and Mitigation Principle

With the data collected and the appropriate model selected, the SDN controller was first initialized in the "detection" state. In this stage, the controller first collects data on the five chosen features every five seconds. These features are then sent to the ML model, which predicts whether the traffic is normal or an attack. If the traffic looks normal, the ML model forwards it in the correct direction. However, if the traffic is classified as an attack the ML model shuts down the port for 20 s to start the mitigation phase. During the mitigation phase, the SDN controller checks the source MAC table to see whether the

MAC address has been learned for a legitimate user, which requires accepting the traffic. Otherwise, the SDN controller does not allow unacceptable traffic. Figure 15 shows how detection and mitigation work in the proposed environment.

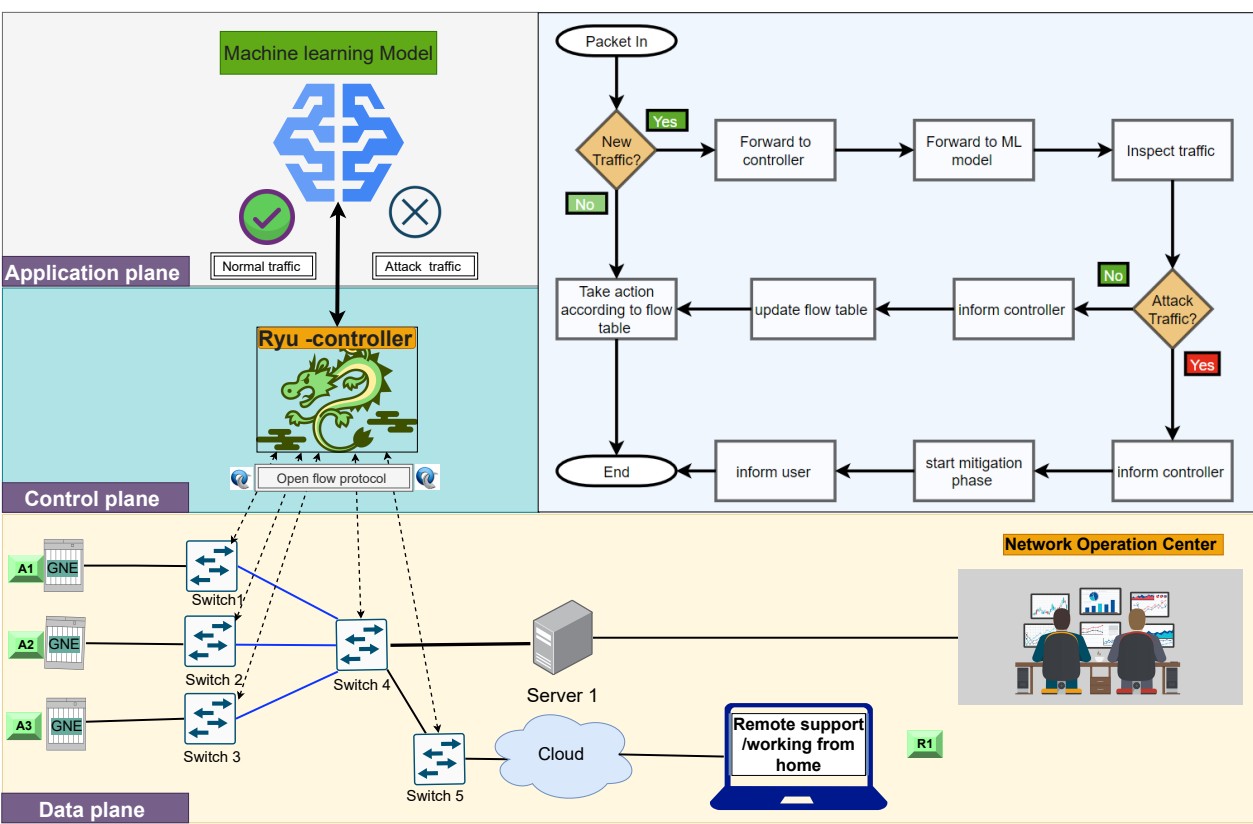

**Figure 15.** Detection and mitigation principle.

### 6.4. Performance Evaluation of FON Using the Selected Model

In this section, we investigate the impact of the proposed model on the primary function of the NMS by targeting two significant network metrics, throughput and delay. To study the impact of DDoS attacks on the throughput and delay, we generated normal TCP traffic from all areas to the server using the Iperf tool, as shown in Figure 15. We used the TCP dump tool to list and collect the incoming traffic on the server side. We supposed a DDoS attack was coming from R1 side to the server using the Hping3 command. We used the ping tool from all sources to the server side to measure the delay.

We found both throughput and delay to be significantly impacted. Figure 16a shows that when not detecting and mitigating a DDoS attack at 60 s the attack is able to begin, resulting in all traffic from the R1 side going down. Other traffic is impacted after a time in between 100 to 120 s. Similar to the throughput, the delay from the R1 side to the server increases dramatically due to the DDoS attack, as shown in Figure 16c, and the delay for other traffic increases after detection of the DDoS attack.

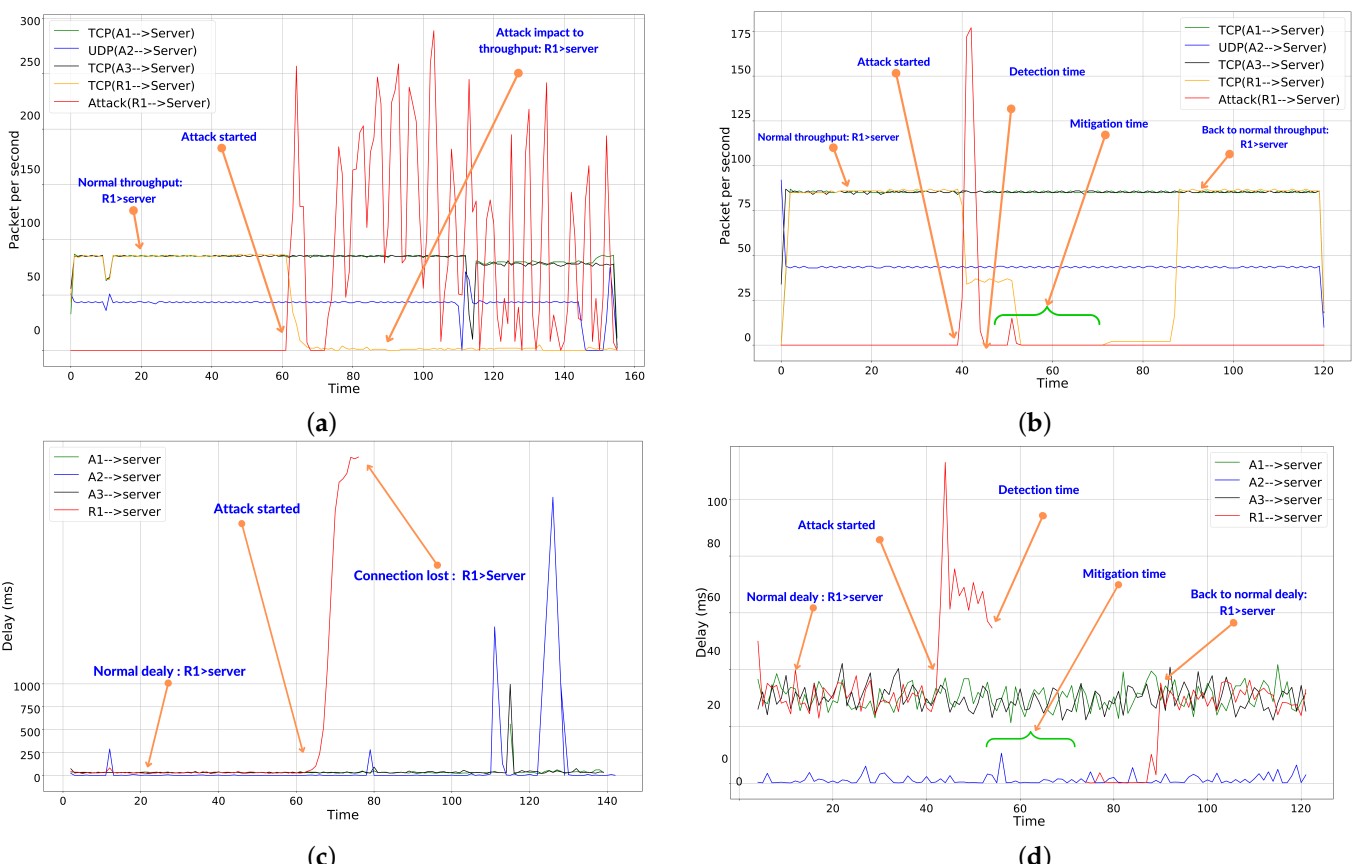

**Figure 16.** Impact of selected model on delay and throughput of FON: (**a**) throughput (without mitigation); (**b**) throughput (with mitigation); (**c**) delay (without mitigation); (**d**) delay (with mitigation).

Our proposed solution can improve the network's throughput and reduce the delay within the DDoS attack period. Considering the example in Figure 16b, where an attack begins after 40 s, it can be seen that the selected RF model first detects the DDoS attack at 45 s and then blocks the source port against incoming attack traffic for 20 s. Second, the mitigation process is immediately activated.

At 70 s, the network returns to its normal state and accepts only traffic from legitimate users. Using an ARP protocol-based solution (port security mechanism), the delay from R1 to the server in Figure 16d increases dramatically compared to that at 40 s. After completing the mitigation phase at 75 s, the delay is reinstated to the normal timing range. Figure 16 shows the impact of our solution on the traffic delay and throughput. These results reflect improvements in the main management function of FON.

### 6.5. Performance Evaluation of the Proposed Model

This section compares the proposed method to the most recent classification-based DDoS detection methods. These baseline approaches are discussed in Section 3. The comparison in this research is based on the number of features, the impact of the introduced new feature (SOS), and how the algorithm collects features to identify a DDoS attack while maintaining high detection accuracy. We selected the two existing approaches in [30] and [27] for use in this performance comparison with our proposed model.

Table 5 shows how the proposed model differs from the above two models discussed in the Related Works section. This analysis justifies the employment of five features, including our new proposed SOS feature, as it can improve performance in terms of accuracy for all selected algorithms. In addition, Figure 17 shows that our proposed solution performs better than the existing related works. However, it is worth mentioning here that our proposed approach can only identify and mitigate DDoS attacks in FONs, and does not

consider malware or malicious traffic. Thus, an additional mechanism addressing this sort of threat could provide more reliable, efficient, and secure data communication, especially in heterogeneous environments. Further statistical testing of our novel approach using sophisticated tool such as analysis of variance (ANOVA) or T-test at various confidence intervals could provide more detailed results as well.

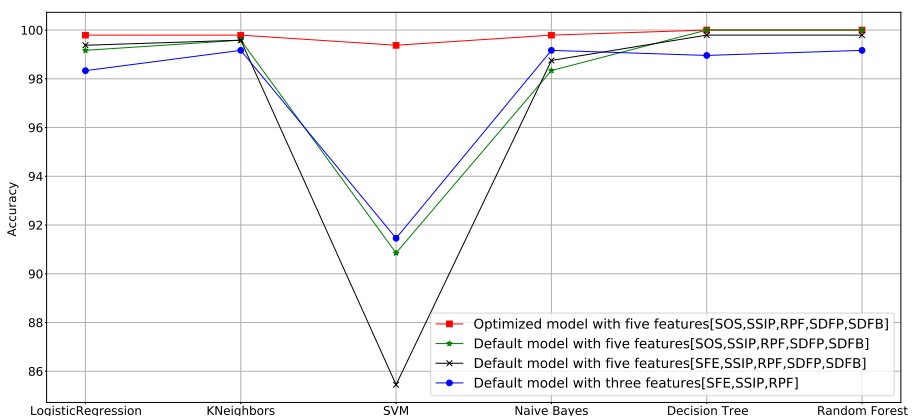

**Figure 17.** Accuracy comparison with existing models (Black [27], Blue [30]).

**Table 5.** Comparison with existing models.

| DDoS D [1] and M [2] Solutions | Isaac et al. [27] | Vishal Kumar [30] | Proposed Model |
|---|---|---|---|
| Features | [SFE, SSIP, RPF, SDFP, SDFB] | [SFE, SSIP, RPF] | [SOS, SSIP, RPF, SDFP, SDFB] |
| Flows in layer [3] | Layer 3 | Layer 3 | Layer 4 |
| Issue | Some features are same | Repeated Instances | - |
| Sample duration | 5 s | 5 s | 5 s |
| Flow counting | Only new flows | Existing flow and new | Only new flows |
| Number of Instances | 1600 | 1600 | 1600 |

[1] Detection. [2] Mitigation. [3] OSI Model.

## 7. Conclusions

This paper presents a study utilizing SDN and ML-based techniques to detect and mitigate DDoS attacks in fiber optic networks. We provide significant technical background on DDoS attack detection and mitigation through various methods published in the literature, then introduce the proposed scheme, tools, and major features used in this research. Based on a performance evaluation, our proposed solution is able to reach an accuracy of 100% using a Random Forest algorithm with five features (SOS, SSIP, RPF, SDFP, SDFB) and a Support Vector Machine algorithm with three features (SOS, SSIP, RPF). We estimate that this result was obtained thanks to the new proposed feature introduced in this paper called Speed of Session (SOS), which is a major contribution of the present research. Furthermore, we explored the effects of this feature through a comparative study with existing solutions in the literature. This paper demonstrates that there can be more than one solution to the problem of DDoS attacks. For example, the fitting time of SVM using three features (SSIP, SOS, and RPF) with C = 1000 and linear Kernal was 14.299 s, and 100% accuracy was reached. On the other hand, when using RF with five features (SSIP, SOS, SDFP, SDFB, and RPF) and default parameters, the accuracy reached 100% and the fitting time was 0.18 s. These results indicate that the Random Forest algorithm is well suited for implementation as it is faster than SVM and uses more affordable features.

**Author Contributions:** Conceptualization, S.A and R.O.; methodology, S.A and R.O.; software, S.A and R.O.; validation, S.A., R.O. and K.S.; formal analysis, S.A., R.O. and K.S.; investigation, S.A., R.O. and K.S.; resources, S.A., R.O. and K.S.; writing—original draft preparation, S.A. and R.O.; writing—review and editing, S.A., R.O. and K.S.; supervision, R.O. and K.S.; project administration,

R.O.; funding acquisition, S.A., R.O. and K.S. All authors have read and agreed to the published version of the manuscript.

**Funding:** The authors extend their appreciation to the Deanship for Research & Innovation, Ministry of Education in Saudi Arabia for funding this research work through the project number IFKSURG-2-13.

**Institutional Review Board Statement:** Not applicable.

**Informed Consent Statement:** Not applicable.

**Data Availability Statement:** Not applicable.

**Acknowledgments:** The authors extend their appreciation to the Deanship for Research & Innovation, Ministry of Education in Saudi Arabia for funding this research work through the project number IFKSURG-2-13.

**Conflicts of Interest:** The authors have no relevant conflict of interest to disclose.

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
