# Peer review of "Using Machine Learning and Software-Defined Networking to Detect and Mitigate DDoS Attacks in Fiber-Optic Networks"

_electronics, doi:10.3390/electronics11234065_

Round 1

Reviewer 1 Report

This manuscript highlights a machine learning-based framework to detect DDoS attacks in Software Defined Networks. The paper is long, and the authors repeat many well-known definitions and concepts. Moreover, several experiments are missing necessary controls. The authors need to improve the overall structure and add relevant details to enrich the quality of the paper. Below are some recommendations to improve the manuscript:

  1. The authors compare six machine learning algorithms. It is unclear why these algorithms are chosen for this analysis. The authors should add relevant references and a paragraph to support the choice of the algorithms.
  2. Section 2, is not a literature review. This section is more of a background to the current research.
  3. FTP, HTTP, SNMP, and ICMP are well known. These protocols do not need definitions
  4. In section 2.5.2, It is unclear why the authors choose SYN, ICMP, UDP, and HTTP flood. Why were Slowris and NTP amplification not considered in this analysis? The authors should add the rationale behind choosing the DDoS scenario.
  5. In the related work section, for each highlighted work, the authors should find gaps and mention how this research will improve the existing research.
  6. In section 4.2, The traffic type needs a better description. The authors should mention the threshold between the “low” and “high” traffic.
  7. For “Speed of the source IP (SSIP)” and “Speed of session (SOS)”, the authors should mention the calculation window.
  8. “Under attack, the number of flow entries at the destination host within the period T increases rapidly, and the destination host cannot respond to them”- This is not always correct for a DDoS attack. The authors should rewrite this statement.
  9. The authors should add an interpretation for figure 9
  10. In figure 10, The authors should highlight the normal and attack time ranges.
  11. In Section 4.33, the description of the machine learning algorithm is not needed. These algorithms are well known.
  12. Figure 12 is not maintaining the aspect ratio.
  13. How did the authors ensure that the algorithms are using the same test train dataset?
  14. Usually, intrusion detection algorithms are optimized to identify False positives. Did the authors optimize the algorithm for false positives? If yes, please mention the methodology.
  15. “Still, the training time is considered high and might not be a good selection in a live detection system.”- This statement is incorrect. Although SVM has a high training time, it is an excellent algorithm for intrusion detection. SVM is very efficient during runtime, and it is considered a faster algorithm.
  16. The authors should mention how the training was optimized to prevent overfitting
  17. In figure 16, why it is unclear why the ML algorithms are trained with limited features.
  18. Unclear sentences:
    1. “With the help of Artificial Intelligent (AI), machine learning lets machines do their jobs skillfully”
    2. “As a result, the Rahman et al. outcomes showed that J48 performs more pleasingly than the other evaluated ML algorithms, especially in training and testing time.”
    3. “Where packeti is the number of packets in the ith flow and Mean_Packets is the average number of packets in the T period in the network. Because of the substantial correlation between this feature and an attack, the standard deviation will be lower for attack data packets than for normal data packets.”

Author Response

Dear Reviewer,

thank you very much for your significant comments that really improved the quality of the work.

best regards

Reviewer 2 Report

This paper presents a study on utilizing SDN and ML-based techniques to detect and mitigate DDoS attacks in fiber-optic network and evaluates six machine learning models and provide good comparison between related works. This paper can be accepted after minor modifications.

1-The manuscript should be written in the Journal Template

2- Kindly avoid using “I” and “We” in the manuscript  and provide passive voice sentences in the manuscript for example:

In conclusion mentioned that “ I provided a significant technical background…”,

                                                      “Table 5 shows how our model differs from ..”

3- In Figs 12-16  and Figs 18—19 written text and the legends are not readable in printed version and 100% zoom.

4-Fig.8 has low quality, which should be modified.

5-Step 3 (The ratio of pair-flow entries (RPF)), in section 4.3. Data collection and feature extraction /selection is exactly copied from “norma.ncirl.ie” , which should be modified.

6- Provide citations for equations, which are not extracted by the authors.

7-Provide some quantitative results with numbers in the abstract.

Author Response

(The authors gave the same response as above.)

Reviewer 3 Report

Using Machine Learning and Software-Defined Networking to
Detect and Mitigate DDoS Attacks in Fiber-Optic Networks

This research used traditional machine learning techniques to identify and counteract distributed denial of service attacks in fiber optic networks. They cite SVM with three features and Random forest with five features as two solutions that have promise for preventing and detecting distributed denial of service attacks in fiber optic networks.The paper is well-written and structured well. All figures are in good form, and the article is written in good condition.

The work is worthy, and I think they should improve it with my below suggestions:

I have major queries :

1.  In the abstract, they should mention which model outperformed and why to detect DDOS in FON? Additionally, they should write the significant vital features too.

2.  You have to need to merge two sections Literature Review & Related work. Or Literature Review can be added in the Introduction section as it is just a basic introduction of concepts.

3.  After the Related work and before the research methodology section, please add a new section, research gap and motivation, you can compare your contribution with Table 1. Add somewhere about your work like: Scope,     Dataset,  features, and Accuracy. Please add your contribution and compare it with existing results.

4.  Possible to compare all algorithm performances with statistical tests such as ANOVA and t-test at various confidence intervals. It makes your results more authenticated.

5.  Before the conclusion, please write the limitations of your work.

Author Response

(The authors gave the same response as above.)

Round 2

Reviewer 3 Report

Dear Authors,

Thanks for answering my queries.